# Boosting Adam-like Optimizers with Signal-to-Noise Ratio Guided Updates

## Abstract

The Adam optimizer remains the default choice in deep learning, offering reliable performance across diverse architectures and tasks. In this work, we reinterpret Adam from a signal-processing perspective—viewing its gradient update as a momentum estimate normalized by noise amplitude—and propose a simple modification: replacing the second raw moment with the second central moment (variance). We show that centering provides a more accurate estimate of noise amplitude, allowing the optimizer to normalize the impact of gradient noise uniformly across the loss landscape and to dynamically scale momentum elements according to their signal-to-noise ratio. Empirically, this modification yields consistent performance gains over Adam and its variants across multiple learning paradigms and neural network architectures, including reinforcement learning and sequence modeling. Notably, on reinforcement learning benchmarks such as MuJoCo, our centered variant called "Adam+" achieves faster convergence and improved stability compared to Adam, which remains the gold standard in settings characterized by non-stationarity and the absence of reliable learning rate schedules.

## 1 Introduction

Adam (Kingma & Ba, 2015) is the de facto default optimization algorithm for modern deep learning. Its success is largely attributed to the combination of momentum (Rumelhart et al., 1986) with adaptive learning rates (Duchi et al., 2011; Tieleman & Hinton, 2012), yielding a robust and memory-efficient method with fast convergence in many deep learning applications. For a parameter $\theta_t$, learning rate $\alpha_t$, bias-corrected first- and second order gradient moments $m_t$ and $v_t$ and a numerical stability constant $\epsilon$, its update rule at iteration $t$ is given by

$$\theta_{t+1} \leftarrow \theta_t - \alpha_t m_t / (\sqrt{v_t} + \epsilon). \tag{1}$$

Many variants of the Adam optimizer have been developed primarily within supervised learning contexts, particularly image classification, to enhance the generalization performance of adaptive gradient methods (Wilson et al., 2017). To mitigate the adverse effects of extreme and unstable per-element learning rate scaling in Adam, AMSGrad (Reddi et al., 2018) and AdaBound (Luo et al., 2019) limit the scaling factor of the learning rate $(\sqrt{v_t} + \epsilon)^{-1}$ in (1). To mitigate the adverse correlation between $m_t$ and $v_t$ in Adam, AdaShift (Zhou et al., 2019) uses delayed gradients for estimating $v_t$, while ADOPT (Taniguchi et al., 2024) reduces this correlation by reordering the updates of $m_t$ and $v_t$. Other notable enhancements include decoupled weight decay introduced in AdamW (Loshchilov & Hutter, 2019), and layer-wise learning rate scaling in LAMB (You et al., 2020) to address the exploding/vanishing gradient problem in very deep architectures.

We revisit Adam from the standpoint of gradient signal-to-noise ratio (SNR). In mini-batch SGD (Robbins & Monro, 1951), gradient estimates contain both signal and noise components, yet most adaptive optimizers—including Adam—use the raw second moment $v_t$ for scaling, conflating variance with mean magnitude. We reinterpret Adam's update (1) as normalizing momentum $m_t$ by an estimate of noise amplitude. From this perspective, Adam can be viewed as approximately scaling updates by SNR, though with estimation error due to its reliance on the raw second moment.

This reinterpretation naturally aligns with earlier work on RMSProp (Tieleman & Hinton, 2012). RMSProp stabilized Adagrad (Duchi et al., 2011) by using an exponential moving average of squared

**Algorithm 1** General optimizer framework

---

**Require:** $\{\alpha_t\}_{t=1}^T, \{\phi_t, \psi_t, \zeta_t\}_{t=1}^T$

1: Initialize $\theta_0$
2: **for** $t = 1$ to $T$ **do**
3: $\quad g_t = \widehat{\nabla J_t}(\theta_t)$
4: $\quad m_t = \phi_t(g_1, \ldots, g_t)$
5: $\quad v_t = \psi_t(g_1, \ldots, g_t)$
6: $\quad \gamma_t = \zeta_t(m_t, v_t)$
7: $\quad \theta_t = \theta_{t-1} - \alpha_t \gamma_t$
8: **end for**

---

Table 1: Functions for Adam in Algo. 1

| | |
|---|---|
| $\phi_t$ | $m_t = \beta_1 m_{t-1} + (1 - \beta_1)g_t$ |
| $\psi_t$ | $v_t = \beta_2 v_{t-1} + (1 - \beta_2)g_t^2$ |
| $\zeta_t$ | $\gamma_t = m_t/(\sqrt{v_t} + \epsilon)$ |

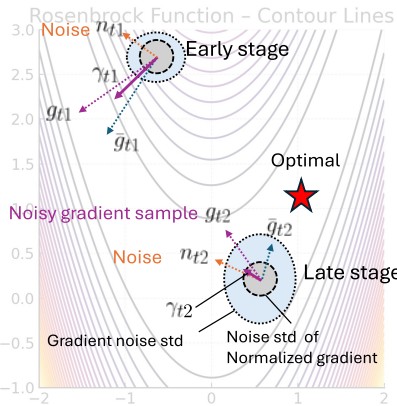

Figure 1: Normalizing noisy gradient sample by noise standard deviation in a 2D plane, $\beta_1 = 0$. In Adam+, each update is normalized by the noise standard deviation, ensuring that updates get smaller as the relative noise level increases in later stages.

gradients. A *centered* variant (Graves, 2013) further subtracted the squared mean gradient. Our work can be seen as bringing this centering principle into Adam's framework. By replacing the raw second moment with the central second moment (variance), we obtain updates directly proportional to gradient SNR. This modification, which we denote Adam+, inherits the efficiency of Adam while improving robustness across diverse and non-stationary settings such as reinforcement learning.

**Contributions:**

- We reinterpret Adam's update rule through the lens of gradient signal-to-noise ratio (SNR), offering a new insight into its behavior.

- Motivated by this SNR-perspective, we introduce a principled enhancement to Adam and Adam-type optimizers by replacing the second raw moment with the central second moment (the variance). The centering of the second raw moment brings consistent improvement across a range of sequential modeling tasks.

- We demonstrate the effectiveness of our approach through extensive empirical evaluations across a range of tasks exhibiting diverse gradient dynamics and optimization algorithms extending beyond Adam-type methods.

## 2 ADAM OPTIMIZER FROM AN INFORMATION THEORETICAL PERSPECTIVE

### 2.1 GENERIC FRAMEWORK FOR OPTIMIZATION ALGORITHMS

Many adaptive optimizers have a unified structure outlined in Algorithm 1. This generic framework (Reddi et al., 2018; Luo et al., 2019) provides a useful lens through which similarities and differences between various algorithms can be analyzed.

In this formulation, $g_t$ is a stochastic gradient sample of the loss function $J_t$ evaluated at time step $t$. The variables $\theta_t$ and $\alpha_t$ represent learnable parameters and learning rates, respectively. The functions $\phi_t$ and $\psi_t$ denote the update rules for the first- and second-order gradient moment estimates, respectively. Often, they incorporate exponential moving averages and bias-correction terms. The function $\zeta_t$ performs normalization and scaling of the gradient moments, and may optionally include additional mechanisms such as decoupled weight decay (Loshchilov & Hutter, 2019). To illustrate how specific optimizers fit into this general framework, we instantiate the $\phi_t$, $\psi_t$, and $\zeta_t$ components for the Adam algorithm (Kingma & Ba, 2015) in Table 1. This shows how Adam performs moment estimation and adaptive per-parameter learning rate scaling.

## 2.2 SNR-BASED REINTERPRETATION OF ADAM-LIKE UPDATES

In the popular mini-batch stochastic gradient descent, the gradient sample can be modeled as

$$g_t = \bar{g}_t + n_t, \tag{2}$$

where $\bar{g}_t$ is the true gradient at time $t$ and $n_t$ is a zero-mean random noise. In a stationary setting, i.e., the distributions of $g_t$ and $n_t$ are time-invariant, the mean and variance of sample $g_t$ are respectively the unbiased estimates of the true gradient and noise power,

$$\bar{g}_t = \mathbb{E}[g_t], \quad \mathbb{E}[n_t^2] = \text{Var}[g_t] = \mathbb{E}[(g_t - \mathbb{E}_t[g_t])^2] . \tag{3}$$

The expected mean and variance can be sampled and estimated, e.g., via a simple moving average (SMA) window. The setting of gradient descent is typically semi-stationary due to slowly changing parameters of neural networks $\theta_t$. In this case, one can replace simple moving average of a window with exponential moving average (EMA), to weigh on recent samples and save memory space for storing gradient samples via the following recursive form, where $0 < \beta < 1$:

$$\bar{g}_t \approx w_t = \beta w_{t-1} + (1 - \beta)g_t, \quad \mathbb{E}[n_t^2] \approx v_t = \beta v_{t-1} + (1 - \beta)(g_t - w_t)^2 . \tag{4}$$

By normalizing the gradient sample $g_t$ or its momentum $m_t$ with the standard deviation of noise $n_t$ in a given local environment, we ensure each step of parameter update $\gamma_t$ contains the same amount of noise across the entire loss landscape. In the minimal example illustrated in Fig. 1, with two points at $t_1$ early stage and $t_2$ late stage of stochastic gradient descent, the raw noisy gradient samples are indicated by dotted arrows, and the noise standard deviation is illustrated by light blue oval shapes. The element-wise normalization changes both the amplitude and direction of both $\gamma_{t1}$ and $\gamma_{t2}$, leading to larger steps in regions with clear gradient, such as early stage of training, and smaller steps in regions that are more noisy, such as flat basin near the optimal point. This behavior is desirable and can accelerate convergence by adjusting the step size based on element-wise gradient SNR, estimated as $w_t^2/v_t$.

In Adam, the second raw moment $v_t$ computed in Table 1 can be interpreted as a biased estimation of the variance in (4), i.e., $\mathbb{E}\left[g_t^2\right] = \text{Var}[g_t] + \mathbb{E}\left[g_t\right]^2$. However, as pointed out in (Zhou et al., 2019), the correlation between $\mathbb{E}\left[g_t^2\right]$ and $\mathbb{E}\left[g_t\right]$ can bring adverse effects, in particular, in regions with strong gradient signal, e.g., when $\mathbb{E}\left[g_t\right]$ is large, the overestimated noise power can overly attenuate high-quality $g_t$ or $m_t$ with large SNR, causing unnecessary slowdown. In dynamic and non-stationary environments like RL, where the errors compound, this could significantly slow down the training process. Centering the second moment, i.e., normalizing updates with the variance of the gradients $\text{Var}[g_t] = \mathbb{E}[g_t^2] - \mathbb{E}[g_t]^2$, helps to reduce the slowdown that occurs in regions with a strong gradient signal. Indeed, centering has shown improvements in RMSProp (Graves, 2013). However, Adam and its descendants inherit the uncentered formulation of RMSProp, and thus the same limitation. Here we present a centered Adam (which we term Adam+) that provides a natural and largely unexplored complement to the Adam family.

When the second raw or central moment becomes very small, the scaling factor $\zeta_t$ may produce overly large effective learning rates, causing instability (Luo et al., 2019). Raw moments tend to overestimate variance, yielding smaller and more stable steps, particularly late in training. In contrast, central moments provide a more accurate SNR estimate but are more sensitive in low-variance regimes. This instability can be alleviated by maintaining a noise floor $\epsilon$ as is done in Adam and Adam-type optimizers.

## 3 ADAM+: ADAM WITH CENTERED SECOND ORDER MOMENT

Based on the generic framework in Algorithm 1 and signal processing interpretation of Adam in section 2.2, we propose to modify the function $\psi_t$.[1]

We propose two key modifications to the core function of Adam-like optimizers $\psi_t$ in Table 1. First, we propose replacing the second raw moment with the second central moment in $\psi_t$,

$$w_t = \beta_2 w_{t-1} + (1 - \beta_2)g_t, \quad v_t = \beta_2 v_{t-1} + (1 - \beta_2)(g_t - w_t)^2 . \tag{5}$$

---

[1] Additionally, two proposed non-linear scaling functions for $\zeta_t$, tailored for supervised learning, are described in the Appendix due to space constraints.

---

**Algorithm 2** Adam+ (modifications with respect to Adam in blue)

---

**Require:** $\alpha_0, \beta_1, \beta_2, \epsilon, \sigma$
1: $m_0 \leftarrow 0, v_0 \leftarrow 0, w_0 \leftarrow 0, t \leftarrow 0$
2: **for** $t = 1$ to $T$ **do**
3: $\quad \xi_t \sim \mathcal{N}(0, \sigma^2)$ /* Optional noise injection */
4: $\quad g_t \leftarrow \nabla_\theta J(\theta_{t-1}) + \xi_t$
5: $\quad m_t \leftarrow \beta_1 m_{t-1} + (1 - \beta_1) g_t$
6: $\quad w_t \leftarrow \beta_2 w_{t-1} + (1 - \beta_2) g_t$
7: $\quad v_t \leftarrow \beta_2 v_{t-1} + (1 - \beta_2)(g_t - w_t)^2$
8: $\quad \hat{m}_t \leftarrow m_t / (1 - \beta_1^t)$
9: $\quad \hat{v}_t \leftarrow v_t / (1 - \beta_2^t)$
10: $\quad \gamma_t \leftarrow \hat{m}_t / (\sqrt{\hat{v}_t} + \epsilon)$
11: $\quad \theta_t \leftarrow \theta_{t-1} - \alpha \gamma_t$
12: **end for**
13: **return** $\theta_t$

---

Notice that in (5), both $w_t$ and $v_t$ are based on the same $\beta_2$, making our approach distinct from a structurally similar update rule in AdaBelief (Zhuang et al., 2020). In AdaBelief, $v_t = \beta_2 v_{t-1} + (1 - \beta_2)(g_t - m_t)^2$, and the $1/\sqrt{v_t}$ is interpreted as a "belief", where $m_t$ is based on the $\phi_t$ in Table 1, which uses $\beta_1$ as the smoothing factor of EMA. While this formulation shares the use of a central moment, its interpretation differs fundamentally from the SNR perspective presented in this work. Our approach introduces a slow momentum $w_t$ for computing the second central moment, faithfully following the EMA approximations of signal and noise power in (4).

Second, we add noise injection $\xi_t$ to the gradient sample as a regularization to reduce the correlation between the fast momentum $m_t$ and the estimation of the noise standard deviation $\sqrt{v_t}$.

We denote Adam with these two modifications as *Adam+*, of which the pseudo code is detailed in Algorithm 2, with our modifications highlighted in blue.

Notice that these key modifications introduced by (5), (including those in the Appendix: (6) and (7)) are also applicable to other Adam-like algorithms, such as AMSGrad, ADOPT, AdamW, and LAMB.

## 4 NUMERICAL RESULTS

To demonstrate that replacing the second raw moment with the variance consistently yields performance gains over the baseline optimizers, we evaluate modified and baseline optimizers on ML tasks that span diverse model architectures, problems, and gradient regimes. [2] [3]

### 4.1 REINFORCEMENT LEARNING

We benchmark the performance of Adam+, with and without noise injection (NI), against the standard Adam optimizer across three MuJoCo environments. Notice that we have used the same set of tuned hyperparameters (Haarnoja et al., 2018; Raffin, 2020) for all simulations. The continuous control tasks feature unbounded return, making them particularly well-suited for evaluating the advantages of improved gradient signal-to-noise handling.

The Fig. 2(a) demonstrates more stable, faster convergence, and higher final performance of Adam+ with and without noise injection over Adam. This indicates that in moderately difficult environments, such as HalfCheetah, the variance normalization helps stabilize early training, and noise injection

---

[2]Source code and data: `https://github.com/researcherAdamPlus/AdamPlus`

[3]All experiments were conducted on a workstation equipped with an AMD Ryzen Threadripper 2970WX 24-core processor (48 threads), 96 GB of RAM, and two NVIDIA GeForce RTX 2080 Ti GPUs. To generate a single representative figure for each task, the computational time was approximately 10 hours for image classification (using either MLP or CNN) and CartPole, 20 hours for the ZINC GT experiment, and 36 hours for training a single agent in the MuJoCo environment.

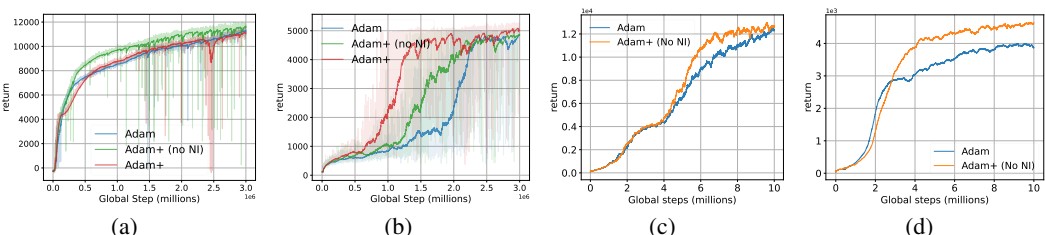

Figure 2: Performance of Adam and Adam+ on MuJoCo and Atari environments. (a) HalfCheetah-v5, (b) Humanoid-v5, (c) Qbert, (d) Seaquest.

may be neutral or slightly detrimental, as excessive randomness can interfere with stable learning once a good policy is found.

Fig. 2(b) highlights the importance of noise injection. For the difficult problem of Humanoid, Adam+ with NI exhibits much faster learning and higher final returns. Adam+ (no NI) performs better than Adam, but lags behind Adam+. This behavior indicates that gradient noise injection provides strong exploration incentives, helping avoid local minima and supporting long-term credit assignment. The combination of variance-aware updates and noise leads to better robustness and learning efficiency.

In addition, we evaluate the performance of our modification on Seaquest and Qbert Atari games solved with DQN. Similar to the continuous control, these games were chosen due to their practically unlimited rewards, which guide the optimizer toward policy improvement. We averaged the results over 5 random seeds. Figs. 2(c) and 2(d) demonstrate that Adam+ consistently outperforms Adam in both convergence speed and final return. In particular, Adam+ reaches high-return regimes significantly earlier, demonstrating improved sample efficiency. For example, in the Seaquest environment, Adam+ attains near-saturated returns after $\sim$4M steps, whereas Adam requires almost twice as many interactions to approach a similar level. This is consistent with our interpretation: by normalizing updates with a more accurate noise variance estimate, Adam+ avoids over-attenuation of strong gradient signals, enabling faster policy improvement.

## 4.2 SEQUENCE MODELING PROBLEMS

Centered RMSProp (Graves, 2013; Ida et al., 2016) tends to perform better on sequence modeling problems compared to its uncentered counterpart. In this subsection, we demonstrate that the centering of Adam has a similar effect. AdaBelief (Zhuang et al., 2020) made an important step in this direction. Building on that, we show that AdaBelief's result can be advanced by ensuring that the variance is estimated with the reference mean of the same window length $\beta_2$. In addition, we demonstrate the performance of our extensions on larger models, such as nanoGPT (Karpathy, 2022) and crammed BERT (Geiping & Goldstein, 2023).

**LSTM on language modeling.** We benchmark Adam+ against AdaBelief on the Penn Treebank (Marcus et al., 1993) language modeling task. For this experiment, we directly reused the publicly released AdaBelief's implementation of this experiment. We reused the learning rate of 0.01 with $\epsilon = 10^{-12}$ and the remaining hyperparameters as in their implementation.

We trained 1-, 2-, and 3-layer LSTMs, averaging results over 5 seeds. Figure 3(a) shows the training trajectory for the 1-layer LSTM; the curves for deeper networks are qualitatively similar. Table 2 summarizes the best training and validation perplexities. Importantly, AdaBelief (Zhuang et al., 2020) has already been shown to outperform several widely used optimizers under the same settings. Since Adam+ achieves better results than AdaBelief with no additional tuning, it follows that Adam+ also surpasses those baselines.

Notice that since AdaBelief estimates the noise variance $v_t$ around $m_t$, its estimate of $\text{Var}(g_t)$ is generally less accurate compared to our case. By contrast, we center $v_t$ around a reference mean within a window of the same length $w_t$, yielding a more reliable estimate. Consequently, our optimizer achieves consistently lower perplexity across different numbers of LSTM layers.

Table 2: Perplexity (lower is better) of Adam+ and AdaBelief on Penn Treebank dataset.

| Optimizer | Training | | | Validation | | |
|---|---|---|---|---|---|---|
| | 1 Layer | 2 Layers | 3 Layers | 1 Layer | 2 Layers | 3 Layers |
| AdaBelief | 60.86 | 45.60 | 37.14 | 81.52 | 66.81 | 61.33 |
| Adam+ | **59.99** | **44.87** | **36.62** | **81.22** | **66.33** | **61.08** |

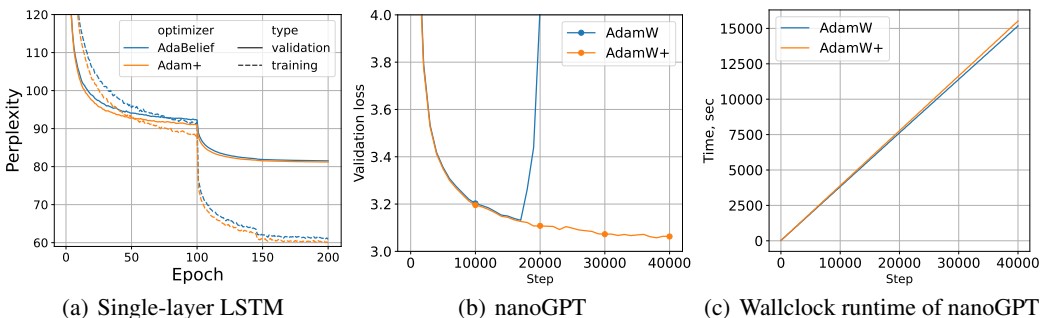

(a) Single-layer LSTM          (b) nanoGPT          (c) Wallclock runtime of nanoGPT

Figure 3: Consistent gain from second moment centering on sequence modeling problems.

**nanoGPT.** We next evaluate the impact of noise centering on nanoGPT (Karpathy, 2022). The experiment is conducted using the default hyperparameters provided in Karpathy (2022). This ensures that any observed improvements are attributable to the optimization mechanism itself, rather than to parameter search. We compare AdamW (Loshchilov & Hutter, 2019) with AdamW+, averaging results over two random seeds. AdamW+ is an optimization algorithm obtained by incorporating the second moment centering from Algorithm 2 into the baseline method of AdamW (Loshchilov & Hutter, 2019).

The outcomes are summarized in Fig. 3(b). Training with AdamW exhibits instability, with validation loss diverging—an issue also documented in the nanoGPT repository (Karpathy, 2022) and in Taniguchi et al. (2024). In contrast, AdamW+ stabilizes training by centering the second raw moment and further achieves consistently lower validation loss relative to AdamW across seeds. As shown in the Fig 3(c), higher stability and lower loss come at the cost of slightly larger runtime due to the extra parameter $w_t$.

**Crammed BERT.** We evaluate the performance of our proposed modifications on fine-tuning BERT (Devlin et al., 2019), following a methodology from Geiping & Goldstein (2023). The model is pretrained with a standard AdamW (Loshchilov & Hutter, 2019) optimizer and is fine-tuned for 5 epochs on a range of downstream tasks given by the GLUE benchmark (Wang et al., 2018). As before, we reuse the tuned hyperparameters from AdamW (Geiping & Goldstein, 2023) and apply them to AdamW+. All results are averaged over five random seeds and reported in Table 3.

Table 3: GLUE benchmark results for crammed BERT averaged over 5 seeds.

| Optimizer | GLUE | | CoLA | MNLI | | MRPC | | QNLI | QQP | | RTE | SST-2 | STS-B | |
|---|---|---|---|---|---|---|---|---|---|---|---|---|---|---|
| | A-Mean | H-Mean | Matthews | Acc | Acc (extra) | Acc | F1 | Acc | Acc | F1 | Acc | Acc | Pearson | Spearman |
| AdamW | 0.8043 | 0.7680 | 0.5024 | 0.8498 | 0.8526 | 0.8252 | 0.8826 | 0.9092 | 0.9082 | 0.8772 | 0.5608 | 0.9323 | 0.8713 | 0.8675 |
| AdamW+ | 0.8061 | 0.7712 | **0.5139** | 0.8484 | 0.8522 | **0.8415** | **0.8926** | 0.9076 | 0.9080 | 0.8769 | 0.5602 | 0.9316 | 0.8719 | 0.8679 |

The arithmetic and harmonic means (A-Mean and H-Mean) indicate that AdamW+ achieves higher performance than AdamW on average across tasks in the GLUE benchmark. Notice that on CoLA and MRPC, AdamW+ brings improvements of at least one percent in Matthew's correlation, accuracy, and F1 score. These cases are bolded in the table. In contrast, for the specific cases where AdamW outperforms AdamW+, the margins are much smaller (strictly less than one percent).

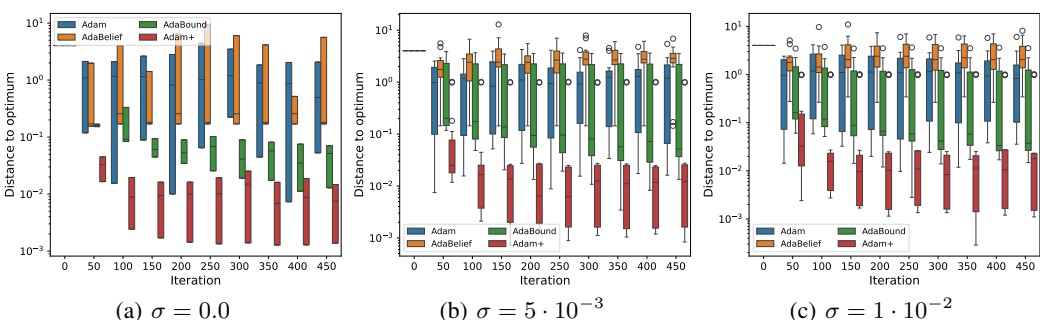

(a) $\sigma = 0.0$      (b) $\sigma = 5 \cdot 10^{-3}$      (c) $\sigma = 1 \cdot 10^{-2}$

Figure 4: Convergence of the proposed Adam extensions on the Rastrigin optimization function for different levels of gradient noise $\sigma$.

### 4.3 RASTRIGIN TEST OPTIMIZATION FUNCTION

We evaluate how different optimizers converge to the optimum in a simple two-dimensional setting given by the Rastrigin test optimization function (Rastrigin, 1974). This setup allows us to compare the convergence speed and the stability of different optimizers on the same loss landscape. We perform a hyperparameter search using the Tree-structured Parzen estimator (TPE) (Watanabe, 2023) for each random seed: decay factors are sampled uniformly from the range $[0.5, 0.999]$, and the learning rates are drawn from a log-uniform distribution over the interval $[e^{-8}, e^{0.5}]$. The number of TPE steps is capped at 500.

To mimic the effect of stochastic data sampling, Gaussian noise $\xi \sim \mathcal{N}(0, \sigma^2)$ is added to the gradients at each iteration, resulting in stochastic optimization trajectories. For statistical significance, we compute 5 trajectories per seed, repeating the procedure over 3 random seeds. We then sample distances to the optimum at selected checkpoints and visualize the distributions using box plots.

The results in Fig. 4 show that Adam+ consistently converges closer to the optimum across all noise regimes, from deterministic gradients ($\sigma = 0$) to high-noise settings ($\sigma = 10^{-2}$). Unlike Adam and AdaBelief, which frequently converge to local minima, Adam+ maintains steady progress toward the global optimum. Moreover, its variance across trials is much smaller (notice the logarithmic scaling of the y-axis), indicating improved robustness to both hyperparameter choices and stochastic perturbations. AdaBound shows partial improvements over Adam but still suffers from wider variability and poorer convergence under higher noise.

These findings highlight two points: (i) centering the second moment reduces the tendency of Adam to over-dampen high-SNR gradients, enabling faster escape from local minima, and (ii) the SNR-based formulation underlying Adam+ provides resilience to injected gradient noise, making its behavior more stable and predictable than other Adam extensions.

### 4.4 MOLECULAR GRAPH REGRESSION

Lastly, we apply our centering extension to a number of different optimizers, where the '+' notation is used to indicate that we incorporate the second moment centering from Algorithm 2 into the baseline method. E.g., LAMB+ corresponds to a centering of the second moment in the baseline algorithm of LAMB (You et al., 2020). We consider the ZINC dataset, comprising approximately $250,000$ molecular graphs with up to 38 atoms (nodes) each, to train the GPS graph transformer (GT) (Rampášek et al., 2022). The task is to regress the continuous molecular properties from the graph structure. The hyperparameters are the same as the original codebase[4], except that we adopt constant learning rates as listed in Table 9.

Figs. 11(a) and 11(b) depict the performance difference in the training and validation mean absolute error (MAE) over the course of the training across a range of $\beta_2$ values. The key observation is that smaller values of $\beta_2$ amplify the gains of the enhanced optimizers. This trend supports the

---

[4]`https://github.com/pyg-team/pytorch_geometric/blob/master/examples/graph_gps.py`

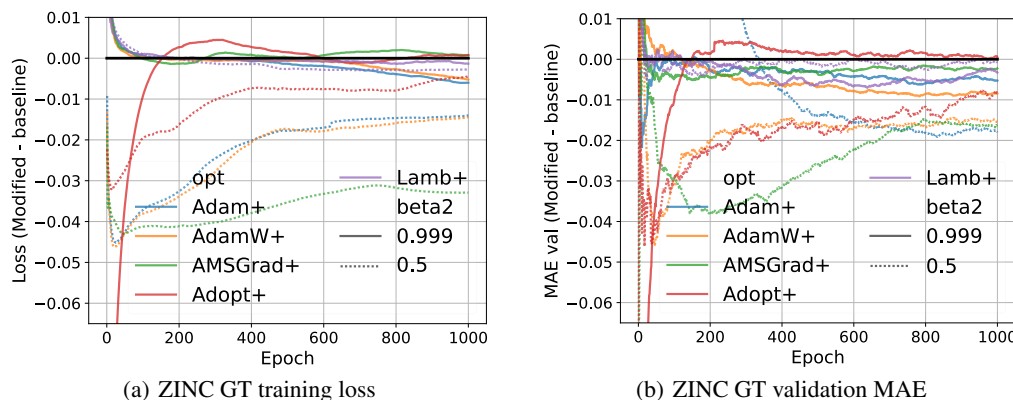

(a) ZINC GT training loss

(b) ZINC GT validation MAE

Figure 5: Performance difference of the modified and baseline optimizers in supervised learning. Molecular graph regression: (a) training loss and (b) validation MAE.

idea that the variance-based moment estimate is more accurate and responsive to recent gradient changes. Adam+, AMSGrad+, and AdamW+ exhibit notable improvements relative to their baselines, which are especially pronounced for lower $\beta_2 = 0.5$. In particular, AMSGrad+ demonstrates substantially faster convergence in the early stages, leading to superior validation MAE in the first half of the training. ADOPT+ also achieves measurable gains, though somewhat smaller, which may be attributed to potential interactions between its moment reordering strategy and the variance-based scaling. In contrast, LAMB+ shows only modest improvements. This is likely due to its trust ratio mechanism, which already stabilizes update magnitudes across layers, reducing the marginal benefit from further normalization via improved noise estimation. More extensive simulation results are summarized in the Tables 8, 9, and 10 in the Appendix.

## 5 RELATED WORK

Despite its popularity, Adam has undergone several refinements aimed at improving its stability and performance. A notable example is AMSGrad (Reddi et al., 2018), which modifies Adam by maintaining the maximum $v_t$ observed so far. Even though this adjustment can lead to overly conservative updates and slower training due to the non-decreasing nature of $v_t$, it has nonetheless inspired a line of research focused on rethinking the mechanisms underlying adaptive optimization.

Luo et al. (2019) argues that the extreme and unstable per-parameter learning rates of Adam contribute to poor generalization (Wilson et al., 2017). In contrast, stochastic gradient descent (SGD) (Robbins & Monro, 1951), though slower to converge, often generalizes better (Luo et al., 2019). To combine the benefits of both methods, the authors proposed AdaBound, an optimizer that constrains the adaptive learning rate within dynamic bounds that gradually tighten toward a fixed target value. This design aims to combine Adam's fast convergence in early training with the generalization benefits of SGD in later stages. Although AdaBound provides a promising trade-off between the adaptability of Adam and the stability of SGD, it requires manual tuning of the bound decay rates. Moreover, the constant terminal learning rate may become suboptimal in non-stationary environments.

AdaBelief (Zhuang et al., 2020) replaces Adam's second moment $v_t$ by the "belief" in the gradient direction $m_t$, $s_t = \beta_2 s_{t-1} + (1 - \beta_2)(g_t - m_t)^2$. Larger "belief" results in a larger per-parameter learning rate. However, this formulation introduces a temporally misaligned update ratio $m_t/(\sqrt{s_t} + \epsilon)$, as $m_t$ and $s_t$ are estimated using different decay factors. Moreover, the notion of "belief" lacks a precise interpretation, limiting its theoretical grounding and connections to related concepts.

Centered RMSProp (Graves, 2013; Ida et al., 2016) uses a temporally aligned update ratio. Specifically, the parameters are updated based on the momentum of rescaled gradients $g_t/(\sqrt{v_t - g_t^2})$. However, this method has two downsides: it relies on the gradient direction given by $g_t$, which is generally less aligned with the true gradient compared to $m_t$, and it lacks bias correction, which can compromise stability in the early stages of training.

Based on the observation that the correlation between $g_t$ and $v_t$ impairs convergence, Zhou et al. (2019) introduce AdaShift: an optimization algorithm which computes $m_t$ from the most recent $n$ gradients, while estimating $v_t$ from a lagged gradient $g_{t-n}$. Although this strategy effectively breaks the correlation (see Theorem 5 in (Zhou et al., 2019)), it is likely to underperform in non-stationary environments, where the gradient dynamics shift over time.

ADOPT (Taniguchi et al., 2024) achieves the same decorrelation effect by excluding $g_t$ from the second moment estimate. Furthermore, the authors identify that Adam-style momentum contributes to convergence issues and address this by reordering the updates of $m_t$ and $v_t$. They prove the convergence to a stationary point for any decay rate factor $\beta_2$.

Recent works extend adaptive methods through multi-scale history or geometric structure. AdaE-MAMix (Pagliarini et al., 2025) stabilizes trajectories using mixtures of fast and slow EMAs. For large-scale pre-training, SOAP (Vyas et al., 2025), Muon (Jordan et al.), and Scion (Pethick et al., 2025) leverage structured preconditioning or tensor orthogonalization to optimize update directions. By contrast, Adam+ maintains the computational efficiency of element-wise updates. Instead of altering the update geometry, Adam+ refines the magnitude of the gradient descent steps by correcting the raw second-order moment estimate via SNR-based centering. This offers a general-purpose improvement without architectural constraints.

Most existing adaptive gradient methods — including Adam and its predecessors Adagrad (Duchi et al., 2011) and uncentered RMSprop (Tieleman & Hinton, 2012) — share a common limitation: they rely on some form of the second *raw* moment of the gradient, $v_t$, to adjust the learning rate. As we illustrated throughout the paper, this reliance restricts their ability to generalize across a wide range of problems, particularly in non-stationary environments, such as RL, where continuous adaptation of the learning rate is critical.

# 6 CONCLUSION

In this work, we interpret the Adam optimization algorithm from the signal-to-noise ratio perspective. This perspective motivates us to replace the raw second moment with the central second moment, yielding Adam+, which normalizes updates by gradient variance and thereby scales gradients in proportion to their SNR.

We empirically validated the effect of our modification across diverse learning paradigms, including reinforcement learning and sequential modeling. These gains are very consistent in sequential modeling problems - the setting where it is known that centered RMSProp outperforms its uncentered counterpart. They are also significant in non-stationary, RL domains, where traditional optimizers often struggle due to unreliable gradient signals and the absence of effective learning rate schedules.

These findings suggest that centering the variance within Adam-like optimizers is a generally useful design principle. Future work may explore integrating this perspective with other recent advances, such as adaptive schedules or large-scale pretraining pipelines, to further enhance the robustness and efficiency of optimization in deep learning.

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

# APPENDIX

## A PROPOSED SNR-BASED MODIFICATIONS TO GRADIENT SCALING FUNCTION $\zeta_t$

Sign-based optimization, such as in Lion (Chen et al., 2023), has strong performance in certain supervised learning applications, as it amplifies small gradient and suppresses strong signals, akin to gradient-guided grid search, which can lead to fast and stable convergence in stationary environments with correct hyperparameters. Inspired by this rationale, we design non-linear scaling function $\psi_t$ on top of Adam+, denoted as "Adam+NLx", by introducing two SNR-based non-linear $\psi_t$ as follows:

$$\gamma_t = \text{sign}(m_t) \log_2 \left( 1 + \sqrt{m_t^2/(v_t + \epsilon)} \right) \tag{6}$$

$$\gamma_t = \text{sign}(m_t) \left[ 1 + \log_{10} \left( 1 + \sqrt{m_t^2/(v_t + \epsilon)} \right) \right] \tag{7}$$

In particular, both (6) and (7) scale the sign update based on element-wise gradient SNR, whereas (6) emphasizes on compressing large SNR for smoother and more robust gradient update, and (7) emulates sign optimizers while scaling the signed momentum by the number of bits in SNR. We denote the optimizers that incorporate this additional refinement with "+NL1" for (6), such as Adam+NL1, and the update in (7) with "+NL2", such as Adam+NL2.

In addition, we consider Adam+(SNR lr) that scales learning rate with SNR as follows:

$$\gamma_t = \sqrt{m_t^2/(v_t + \epsilon)} \cdot m_t/(\sqrt{v_t} + \epsilon) \,, \tag{8}$$

which is evaluated for RL in CartPole experiment.

We propose two more gradient scaling functions $\zeta_t$ in Algorithm 1, and summarize all our enhancements to Adam variants in Table 4, listing their postfixes, equations and algorithms, recommended baselines, hyperparameters, and application scenarios.

Table 4: The full list of our enhancements to Adam variants

| Recommend Baseline | Postfix | Enhancement | | Recommend Hyperparameters | Recommend Applications |
| --- | --- | --- | --- | --- | --- |
| | | $\psi_t$ | $\zeta_t$ | | |
| Adam, | + | (5) | Adam | | General, RL |
| AdamW, | + | (5) | Adam | $\sigma \in \{0.001, 0.0001\}$ | RL |
| LAMB | + (SNR lr) | (5) | (9) | | RL |
| Adam, | +NL1 | (5) | (10) | | Supervised learning |
| AdamW | +NL2 | (5) | (11) | $\sigma = 0$ | Supervised learning |
| | +NL3 | (5) | (12) | | Supervised learning |

We still recommend $\beta_1 = 0.9, \beta_2 = 0.999$ as the default configuration with our enhancements for most application scenarios, however, as shown in Appendix F, smaller $\beta$, such as $\beta_1 = \beta_2 = 0.5$ and $\beta_1 = \beta_2 = 0.9$, with our modifications on Adam, AdamW and Adopt can lead to superior performance compared to the default values for graph transformers.

Next, the proposed non-linear gradient scaling $\zeta_t$ functions in Table 10 are detailed as follows.

**Adam+ (SNR lr)**: Postfix (SNR lr) stands for SNR-based learning rate scaling, expressed as

$$\gamma_t = \sqrt{m_t^2/(v_t + \epsilon)} \cdot m_t/(\sqrt{v_t} + \epsilon) \,. \tag{9}$$

Adam+ (SNR lr) scales the Adam+ update $m_t/(\sqrt{v_t} + \epsilon)$ by the square root of SNR, essentially employing quadratic Adam+ update with the sign of momentum $m_t$. This approach promotes faster responses under high SNR and slower responses in low SNR regime.

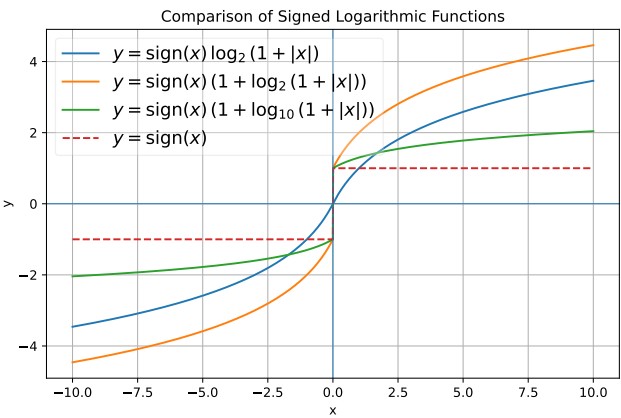

Figure 6: Signed logarithmic functions in (10), (11), (12) for +NL1, +NL2, +NL3

**Adam+NL1~3**: Lastly, signed logarithmic functions, as illustrated in Fig. 6, are employed as an additional non-linear filtering mechanism in our extensions: (10) for *NL1*, (11) for *NL2*, and (12) for *NL3*, where NL1 and NL2 are already introduced in the main text.

$$\gamma_t = \text{sign}(m_t) \log_2 \left( 1 + \sqrt{m_t^2/(v_t + \epsilon)} \right) \tag{10}$$

$$\gamma_t = \text{sign}(m_t) \left[ 1 + \log_{10} \left( 1 + \sqrt{m_t^2/(v_t + \epsilon)} \right) \right] \tag{11}$$

$$\gamma_t = \text{sign}(m_t) \left[ 1 + \log_2 \left( 1 + \sqrt{m_t^2/(v_t + \epsilon)} \right) \right] \tag{12}$$

As shown in the ZINC-GT experiment in Appendix F, the *+NL3* extension achieves the best performance in training graph transformer compared to +NL1 and +NL2.

The rationale behind these logarithmic functions is to dampen strong elements and elevate weak elements in the normalized updates of Adam+ for supervised learning, in which gradient updates become very weak as training progresses, as shown in Figs. 7 in Appendix B. NL1~3 serve as a middle ground between signed optimizers, such as Lion (Chen et al., 2023), and Adam-like linear optimizers. Similar to Lion, NL1~3 are mainly for supervised learning, which underperform linear optimizers in reinforcement learning, where fast responses to disruptions are critical.

## B  GRADIENT SNR IN CIFAR10-RESNET18 AND CARTPOLE-DQN

In this section, we analyze the dynamics of element-wise gradient SNR, a central concept underlying our enhancements to Adam-like baseline optimization algorithms. We trace key metrics for both Adam and Adam+ in the contexts of supervised learning (CIFAR-10 with ResNet-18) in Figs. 7 and reinforcement learning (CartPole with DQN) in Figs. 8. To trace gradient SNR, we add functions for layer-wise gradient SNR estimation to Adam+ and our customized Adam optimizers.

We compute the gradient SNR in dB domain for the last layer of the neural network, as follows:

$$\overline{\eta}_t = \frac{1}{d} \sum_{i=1}^{d} \eta_{t,i} \,, \quad \eta_{t,i} = 10 \log_{10} \left( \frac{m_{t,i}^2}{v_{t,i} + \epsilon} \right) \,,$$

where $\eta_{t,i}$ is the element-wise gradient SNR in dB for element $i$ at time step $t$, and $d$ denotes the number of parameters of the layer. Similarly, we compute layer-wise average second moment as

$$\bar{v}_t = \frac{1}{d}\sum_{i=1}^{d} 10 \log_{10}(v_{t,i} + \epsilon).$$

The reason to average in dB domain is to prevent the gradient SNR or second moment from being dominated by elements that are orders of magnitude larger than the typical values, reflecting the true scale of most elements.

We focus on the last layer for two reasons: (i) it reduces computational costs compared to computing SNR across the entire model, (ii) the last-layer gradient has the most immediate impact on the loss, making it a meaningful indicator of convergence behavior.

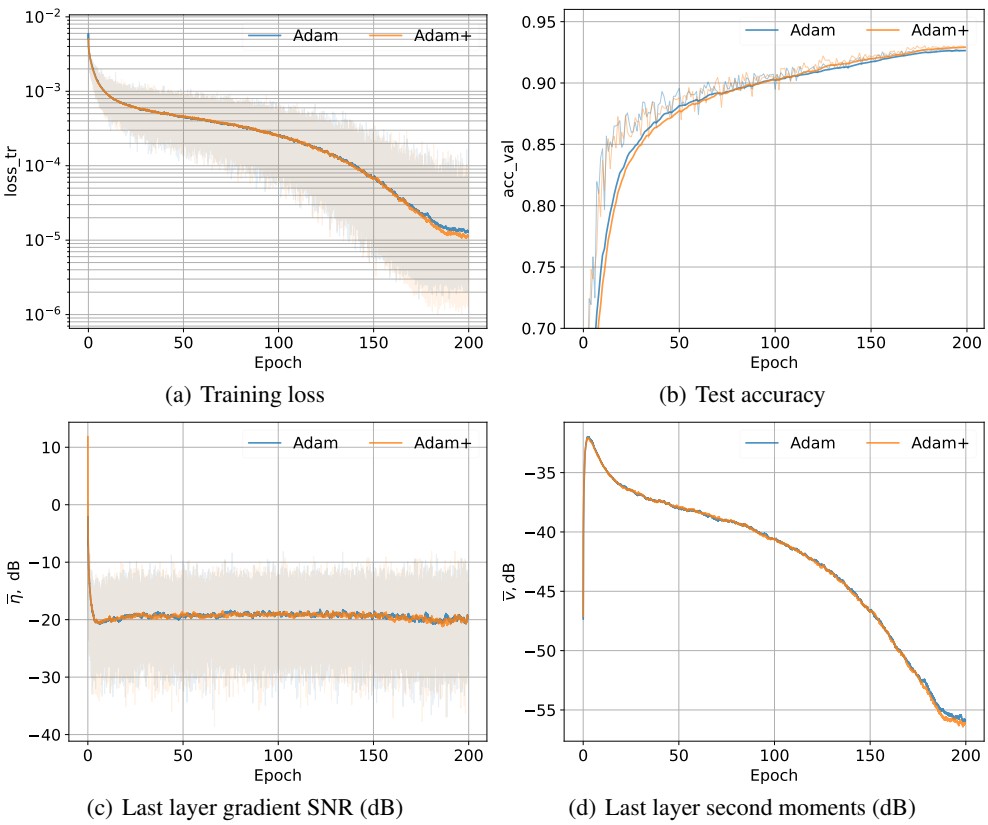

(a) Training loss

(b) Test accuracy

(c) Last layer gradient SNR (dB)

(d) Last layer second moments (dB)

Figure 7: Traces of training ResNet-18 on CIFAR-10 under cosine learning rate annealing: (a) training loss, (b) test accuracy, (c) gradient SNR in dB, and (d) second raw moment for Adam and second central moment for Adam+ in dB. (c) and (d) are only for the last layer. ($\beta_1 = 0.9, \beta_2 = 0.999$).

As shown in Figs. 7, the training loss, test accuracy, gradient SNR and second moments of the last layer in ResNet-18 of Adam and Adam+ are closely aligned over 200 epochs in supervised learning, with Adam+ slightly leads in the last 30 epochs. For most of the time, the gradient SNR of the last layer fluctuate between $-30 \sim -12$ dB, with an average of -20 dB, however, the second moments decrease steadily over the course of training, from $-32$ dB in the early epochs to $-56$ dB in the end, reducing two orders of magnitude. Given the almost constant gradient SNR, this shows that the scale of gradient update decreases by at least an order of magnitude over the course of training. This highlight the challenge of supervised learning in shrinking magnitude of gradient.

The different smoothness of gradient SNR $\bar{\eta}_t$ and second moments $\bar{v}_t$ can be explained by their different smoothing factor, $\beta_1 = 0.9$ for $\bar{\eta}_t$ and $\beta_2 = 0.999$ for $\bar{v}_t$. Toward the end of the training, the second central moment of Adam+ becomes visibly smaller than the second raw moment of Adam,

as shown in Fig. 7(d), leading to larger gradient update, which may contribute to the better training loss and test accuracy of Adam+ compared to Adam.

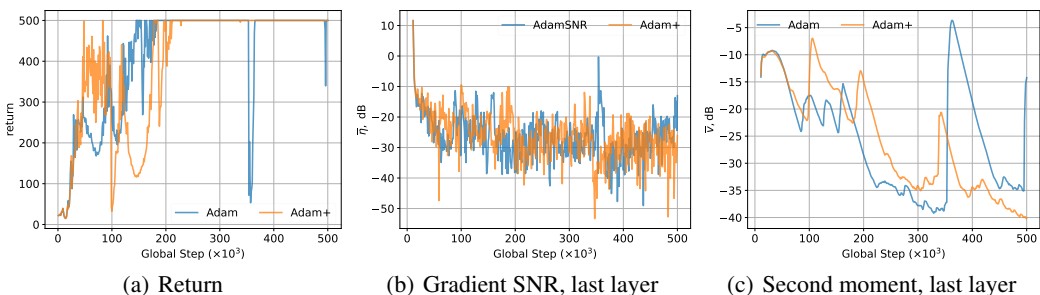

(a) Return           (b) Gradient SNR, last layer           (c) Second moment, last layer

Figure 8: Return and last-layer metrics for DQN in CartPole under Adam and Adam+ in a single run (random seed: 50) with constant learning rate, based on hyperparameters in Table 14.

The return and the gradient SNR and second moments of the last layer for DQN in Cartpole in a single run are presented in Figs. 8. In Fig. 8(a), the DQN initially converges around 200,000 steps under both Adam and Adam+, but subject to disturbances from exploration at ratio of 0.05 (see Table 14), which encourage robustness and generalizability. This explains drops in return after the initial convergence under both Adam and Adam+. However, Adam and Adam+ respond differently to such disturbances: the DQN under Adam+ quickly rebalanced with only small drops in return, but under Adam the drop is more substantial and recovery is slower.

The traces of last layer gradient SNR (Fig. 8(b)) and second moments (Fig. 8(c)) under Adam and Adam+ provide more insights to the optimizer behavior and the environmental dynamics. First, the gradient SNR in RL is much lower and more volatile (range of $-20 \sim -50$ dB with a gradually decreasing mean) than that of supervised learning ($-12 \sim -30$ dB with a constant average of $-20$ dB), indicating a more difficult, non-stationary environment for reinforcement learning.

Second, when disturbance appears, e.g., around 350,000 for Adam, and 340,000 for Adam+, the gradient SNR and second moment jump up, indicating a departure from the convergence area. However, the second moment under Adam raises more substantially (17 dB higher) compared to Adam+ under similar disturbances. Such a smaller increase in second moment under Adam+ allow larger learning rate being applied when the gradient SNR increases abruptly, leading to fast responses to disturbance and only a tiny drop in return. In contrast, the smaller learning rate scale under Adam in such disturbance leads to slower recovery, and substantial drops in return under disturbances. This example demonstrates the benefit of using second central moment instead of second raw moment in Adam+ in reinforcement learning under tough, non-stationary environments.

Table 5: Training hyperparameters for CIFAR-10 with ResNet18

| Hyperparameter | Value |
| --- | --- |
| Dataset | CIFAR-10 (Krizhevsky, 2009) |
| Model | ResNet18 (He et al., 2016) |
| Batch size | 128 |
| Epochs | 200 |
| $\beta_1, \beta_2$ | 0.9, 0.999 (default) |
| Learning rate (initial) | 0.001 |
| Learning rate schedule | Cosine annealing |
| Weight decay | $5 \times 10^{-4}$ |
| Data augmentation | Random crop, horizontal flip |
| Loss function | Cross-entropy |

## C   EXPERIMENTAL SETUPS AND HYPERPARAMETERS FOR EVALUATED PIPELINES

In Table 6, we list the hardware specifications, experimental setups and runtime for our evaluations of various optimizers across different deep learning pipelines, including the code base for the five deep learning pipelines and our modifications. The hyperparameters of each deep learning pipeline are further detailed in Table 5 for CIFAR10-ResNet18, Table 7 for MNIST-MLP, Table 11 for ZINC-GT, Table 14 for CartPole-DQN, and Table 16 for MuJoCo-SAC.

Table 6: Hardware configuration, runtime, and code bases for all pipelines

| Item | Details |
|---|---|
| Hardware specs. | AMD Ryzen Threadripper 2970WX (24 cores / 48 threads), 96 GB RAM, dual NVIDIA RTX 2080 Ti GPUs |
| `CIFAR10-ResNet18` | **Dataset:** CIFAR-10 (Krizhevsky, 2009) 
 **Runtime:** 1.2 hours per optimizer (1 thread per GPU) 
 **Codebase:** `https://github.com/kuangliu/pytorch-cifar` 
 **Notes:** Used standard ResNet18 and cosine LR schedule. |
| `MNIST-MLP` | **Dataset:** MNIST digits (Lecun et al., 1998) 
 **Runtime:** 30 minutes per optimizer (1 thread per GPU) 
 **Codebase:** `https://github.com/tensorflow/datasets/blob/master/docs/keras_example.ipynb`, `https://github.com/pytorch/tutorials/blob/main/beginner_source/blitz/cifar10_tutorial.py` 
 **Notes:** Trained with 2-layer MLP. The pipeline is based on the two referenced codebases. |
| `ZINC-GT` | **Dataset:** ZINC molecular graphs (Gómez-Bombarelli et al., 2018) 
 **Runtime:** $6 \sim 8$ hours per optimizer (10 threads in parallel) 
 **Codebase:** `https://github.com/pyg-team/pytorch_geometric/blob/master/examples/graph_gps.py` 
 **Notes:** Epoch-based random seed for training dataloader, fixed seed for validation and test dataloaders. |
| `CartPole-DQN` | **Environment:** OpenAI Gym CartPole-v1 (Huang et al., 2022; Towers et al., 2024) 
 **Runtime:** 1.2 hours per optimizer (10 seeds in parallel) 
 **Codebase:** `https://github.com/vwxyzjn/cleanrl/blob/master/cleanrl/dqn.py` 
 **Notes:** Discrete control task; all optimizers evaluated in same seed regime ($20 \sim 29$). |
| `MuJoCo-SAC` | **Environment:** MuJoCo continuous control (v5) (Huang et al., 2022; Towers et al., 2024) 
 **Runtime:** 40 hours for 3 million global steps 
 **Codebase:** `https://github.com/vwxyzjn/cleanrl/blob/master/cleanrl/sac_continuous_action.py` 
 **Notes:** Long-horizon RL; unbounded return. |

## D   ADDITIONAL RESULTS ON IMAGE CLASSIFICATION AND RL CONTROL

The results for CIFAR-ResNet18[5], MNIST-MLP, ZINC-GT, and DQN-cartpole are summarized in Table 8. A key observation is that no single optimizer consistently outperforms others on every task, whereas our modifications generally enhance the baseline optimizers across all tested settings.

---

[5]Based on codebase in `https://github.com/kuangliu/pytorch-cifar`. Also, see Table 5 in the Appendix C.

Table 7: Training hyperparameters for MNIST with 2-layer MLP

| Hyperparameter | Value |
|---|---|
| Dataset | MNIST (Lecun et al., 1998) |
| Model | 2-layer MLP (784-128-10) |
| Activation | ReLU |
| Batch size | 100 |
| Epochs | 200 |
| Learning rate | $1 \times 10^{-3}$ for LAMB, LAMB+; $1 \times 10^{-4}$ for the rest |
| Weight decay | $1 \times 10^{-3}$ |
| Dropout | None |
| Loss function | Cross-entropy |

Notice that the goal of these experiments is *NOT* to introduce an optimizer that universally outperforms all others across all tasks, as shown by the test results that none of them could do so.

For each task, all optimizers were tested under identical experimental conditions. The only differences lie in the optimizer configurations, ensuring a fair and controlled comparison. We consider three benchmarks: image classification, and reinforcement learning. We train an image classifier on CIFAR-10 (Krizhevsky, 2009) and MNIST (Lecun et al., 1998) datasets with ResNet-18 (He et al., 2016) and MLP, respectively. Lastly, we evaluate the performance of Adam and Adam+ in CartPole with deep Q network (DQN) (Mnih et al., 2015).

## D.1 IMAGE CLASSIFICATION

**MLP for MNIST** We trained a 2-layer MLP classifier for the MNIST digit classification task. The experimental settings are provided in Table 7 in the Appendix. Table 8 demonstrates a consistent improvement in the maximum validation accuracy resulting from our enhancements. Fig. D.1 demonstrates the difference in validation accuracy between the enhanced and the baseline versions of 5 optimizers over the course of training for $\beta_2 = 0.999$.

Generally, we see a positive trend indicating the consistent gains attained by the "+" versions. Among them, Adam+ demonstrates the most consistent improvements over standard Adam. By decoupling the signal from the noise — through replacing the second raw moment with the gradient variance — Adam+ enables more accurate estimation of the true gradient direction, thereby facilitating improved convergence. Similar in spirit, LAMB+, which scales updates using a trust ratio, also benefits from more accurate SNR estimation. This leads to sustained performance gains across training, highlighting the compatibility of variance-based normalization with layer-wise adaptive scaling. In contrast, the improvement of AMSGrad+ over AMSGrad is more moderate. This is likely due to the inherently conservative nature of both optimizers, as they retain the maximum of previous second moment estimates, thereby reducing the dynamic range available for further enhancement. Finally, AdamW+ and ADOPT+ exhibit more erratic behavior, with gains followed by periods of degradation. This instability may arise because both optimizers are already finely tuned for strong performance on standard image classification tasks, leaving less room for consistent improvements via second-moment modifications.

To quantitatively assess optimizer convergence, we employed two metrics: the normalized area under the curve (nAUC) and normalized full score duration (nFSD). The nAUC measures the agent's cumulative performance over the entire training process and is normalized by the product of the maximum achievable return (500) and the total number of steps. In contrast, the nFSD metric measures the fraction of steps at which the agent achieves maximum return. The normalization coefficient is defined by the total number of steps performed by an agent. Table 8 shows that Adam+ ranks the highest in both measures. Furthermore, the improved optimizers outperform the originals across the board, underscoring the effectiveness of our modifications.

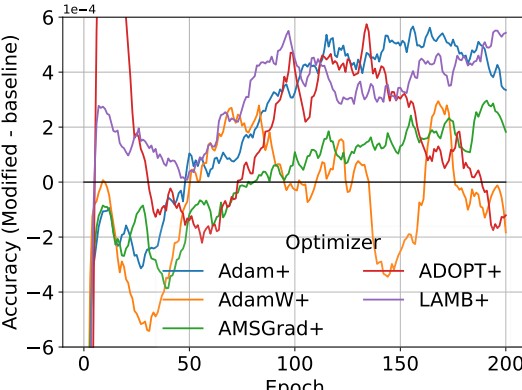

Table 8: Optimizer performances in image classification (test accuracy for CIFAR10-ResNet18 and MNIST-MLP), graph-level regression using a graph transformer in ZINC (loss value, Mean Absolute Error (MAE) in validation and testing), and reinforcement learning using a DQN for cartpole (normalized area under the curve and normalized full score duration).

| Optimizer $\beta_1 =$ 0.9, $\beta_2 = 0.999$ | CIFAR10 | MNIST | ZINC, GT | | | Cartpole | |
|---|---|---|---|---|---|---|---|
| | Accuracy | Accuracy | Loss | MAE val | MAE test | nAUC | nFSD |
| AdamW+NL1 | 0.9422 | 0.9800 | 0.0572 | 0.1555 | 0.1215 | 0.8449 | 0.155 |
| AdamW+ | 0.9398 | 0.9807 | 0.0551 | 0.1603 | 0.1338 | 0.8367 | 0.200 |
| AdamW | 0.9391 | 0.9804 | 0.0616 | 0.1690 | 0.1312 | 0.8225 | 0.205 |
| AMSGrad+ | 0.9388 | 0.9803 | 0.0638 | 0.1663 | 0.1349 | 0.8165 | 0.230 |
| AMSGrad | 0.9406 | 0.9801 | 0.0636 | 0.1684 | 0.1336 | 0.8083 | 0.245 |
| LAMB+ | 0.9364 | **0.9828** | 0.0443 | 0.1230 | 0.0913 | 0.6836 | 0.005 |
| LAMB | 0.9352 | 0.9825 | 0.0463 | 0.1275 | 0.0973 | 0.6336 | 0.000 |
| Adam+ (SNR lr) | - | - | - | - | - | **0.8579** | **0.420** |
| Adam+ | 0.9327 | 0.9815 | 0.0547 | 0.1621 | 0.1332 | 0.8577 | 0.330 |
| Adam | 0.9323 | 0.9811 | 0.0610 | 0.1681 | 0.1329 | 0.8149 | 0.120 |
| ADOPT | 0.9388 | 0.9813 | 0.0548 | 0.1581 | 0.1280 | 0.8263 | 0.160 |
| ADOPT+ | 0.9374 | 0.9814 | 0.0560 | 0.1592 | 0.1339 | 0.8190 | 0.280 |
| AdaBound | **0.9428** | 0.9821 | 0.0882 | 0.1707 | 0.1460 | 0.0322 | 0.000 |
| AdaBeliefW | 0.9402 | 0.9804 | 0.0597 | 0.1639 | 0.1262 | 0.8334 | 0.145 |
| Lion ($\beta_2 = 0.99$) | 0.9372 | 0.9773 | **0.0335** | **0.1090** | **0.0863** | 0.4564 | 0.000 |
| AdaBelief | 0.9326 | 0.9804 | 0.0549 | 0.1583 | 0.1271 | 0.7984 | 0.120 |

## E    EXTENDED RESULTS FOR THE TASKS IN TABLE 8

We further expand the results for optimizers under four deep learning tasks in Table 8 into Table 10, by including the test results of four additional optimizers, PIDAOSI, RMSprop, and AdaShift. In Table 10, the performance metrics of deep learning pipelines using various optimizers are listed, which cover different neural network architectures (CNN, MLP, Transformers), tasks (image classification, graph-level regression, control), and learning paradigms (supervised learning and reinforcement learning). The top three optimizers for each pipeline are highlighted in colors.

As noted in the main body of the paper, image classification is one of the most well-studied tasks, therefore, the differences in performance (test accuracy) under different optimizers are generally very small. Also notice that for ZINC-GT, and Cartpole-DQN, more optimizers and parameters ($\beta_1, \beta_2$) are evaluated in Appendices F and G. For example, the top three for ZINC-GT pipeline in Table 10 are not necessarily the best when different ($\beta_1, \beta_2$) or learning rate schedules are used.

The key messages from Table 10 are as follows:

- None of the optimizers dominate others across all the four tasks. PIDAOSI ranks high across all four tasks, but not in top three for graph level regression. Lion leads in ZINC-GT pipeline by a significant margin, but perform poorly in Cartpole-DQN.

Table 9: Optimizer performances on molecular graph regression with graph transformers.

| $(\beta_1, \beta_2)$ | Optimizer (lr) | Loss | MAE val | MAE test | Optimizer (lr) | Loss | MAE val | MAE test |
|---|---|---|---|---|---|---|---|---|
| 0.5, 0.5 | AdamW+NL2 (0.0001) | **0.0429** | **0.1253** | 0.1008 | Adam+NL2 (0.0001) | 0.0466 | 0.1295 | 0.0981 |
| 0.9, 0.9 | | 0.0433 | 0.1261 | **0.0945** | | **0.0463** | **0.1270** | **0.0970** |
| 0.9, 0.999 | | 0.0435 | 0.1299 | 0.0966 | | 0.0480 | 0.1318 | 0.1017 |
| 0.5, 0.5 | AdamW+NL1 (0.0001) | 0.0459 | 0.1422 | 0.1111 | Adam+NL1 (0.0001) | 0.0472 | 0.1412 | 0.1091 |
| 0.9, 0.9 | | 0.0533 | 0.1569 | 0.1240 | | 0.0514 | 0.1559 | 0.1268 |
| 0.9, 0.999 | | 0.0572 | 0.1555 | 0.1215 | | 0.0581 | 0.1613 | 0.1252 |
| 0.5, 0.5 | AdamW+ (0.0001) | 0.0532 | 0.1520 | 0.1182 | Adam+ (0.0001) | 0.0530 | 0.1534 | 0.1196 |
| 0.9, 0.9 | | 0.0548 | 0.1594 | 0.1273 | | 0.0599 | 0.1645 | 0.1302 |
| 0.9, 0.999 | | 0.0551 | 0.1603 | 0.1338 | | 0.0547 | 0.1621 | 0.1332 |
| 0.5, 0.5 | AdamW (0.0001) | 0.0667 | 0.1662 | 0.1315 | Adam (0.0001) | 0.0673 | 0.1697 | 0.1299 |
| 0.9, 0.9 | | 0.0620 | 0.1706 | 0.1307 | | 0.0620 | 0.1650 | 0.1297 |
| 0.9, 0.999 | | 0.0616 | 0.1690 | 0.1312 | | 0.0610 | 0.1681 | 0.1329 |
| 0.5, 0.5 | LAMB+ (0.0010) | 0.0560 | 0.1479 | 0.1225 | ADOPT+ (0.0001) | **0.0547** | **0.1456** | **0.1174** |
| 0.9, 0.9 | | 0.0466 | 0.1266 | 0.0955 | | 0.0608 | 0.1588 | 0.1292 |
| 0.9, 0.999 | | **0.0443** | **0.1230** | **0.0913** | | 0.0548 | 0.1582 | 0.1281 |
| 0.5, 0.5 | LAMB (0.0010) | 0.0588 | 0.1485 | 0.1207 | ADOPT (0.0001) | 0.0583 | 0.1568 | 0.1297 |
| 0.9, 0.9 | | 0.0467 | 0.1268 | 0.0952 | | 0.0536 | 0.1579 | 0.1281 |
| 0.9, 0.999 | | 0.0463 | 0.1276 | 0.0974 | | 0.0560 | 0.1593 | 0.1339 |
| 0.5, 0.5 | AMSGrad+ (0.0001) | 0.0938 | 0.1989 | 0.1602 | AMSGrad (0.0001) | 0.1268 | 0.2157 | 0.1853 |
| 0.9, 0.9 | | 0.0850 | 0.1867 | 0.1587 | | 0.0879 | 0.1885 | 0.1542 |
| 0.9, 0.999 | | **0.0638** | **0.1663** | **0.1349** | | 0.1718 | 0.2434 | 0.2098 |

- Our modifications generally bring consistent improvements over their baselines. In particular, Adam+, AdamW+, and LAMB+ consistently outperform their baselines Adam, AdamW, and LAMB across the four tasks. For ADOPT and AMSGrad, such benefits are less consistent.

- Our modified optimizers generally ranked among the top three across all pipelines. In Cartpole-DQN, our Adam+, Adam+ (SNR lr) ranks the top within the top two in nAUC and the top two in nFSD, demonstrating the strength and utility of our enhancement in this domain.

We further evaluate optimizers in the ZINC-GT pipeline with different combinations of $(\beta_1, \beta_2)$ and learning rate schedulers in Appendix F. For reinforcement learning, more results are presented in Appendices G and I.

# F  HYPERPARAMETES AND TEST RESULTS FOR GRAPH TRANSFORMER ON ZINC DATASET

In this section, we further analyze the optimizers' performance on the graph regression problem with graph transformers. The hyperparameter settings are given in the Table 11. For a fair evaluation of the optimizers, we replace the *ReduceLROnPlateau* learning rate scheduler with two settings: constant learning rate in Table 12, and cosine learning annealing in Table 13. As a result, our minimum MAE on the test set is higher than that in the original paper (Rampášek et al., 2022) (MAE: 7%), since we do not employ a validation set for learning rate decay. However, in our setting, validation MAE serves as another independent test set since it is not involved in training.

Table 12 serves as an extension of the Table 9 and includes additional optimizers, such as AdamW+NL3, AdaBelief, AdaShift, RMPprop, AdaBound, and PIDAOSI under constant learning rate. In this setting, Lion achieves the best performance by a large margin in all three MAE on the training set, validation set, and test set. Our enhanced LAMB+ achieves the second in validation and test MAE, which is not completely surprising since LAMB is designed for deep architecture like transformers. Lastly, our non-linear modification to AdamW, AdamW+NL3 achieves the third place in validation and test MAE. Moreover, within each comparison group, our enhanced versions constantly outperform their baseline counterparts, highlighting the benefits of our enhancement. It can also be observed that for AdamW and Adam, smaller $\beta$s lead to worse performance, while our

Table 10: Optimizer performances in image classification (CIFAR, MNIST, **larger is better**), molecular graph regression using a graph transformer in ZINC dataset (**smaller is better**) with constant learning rates, and reinforcement learning using a DQN for cartpole (normalized area under the curve and normalized full score duration) (**larger is better**). Highlights are the top first, second, and third.

| Optimizer $\beta_1 = 0.9, \beta_2 = 0.999$ | CIFAR10 | MNIST | ZINC, GT | | | Cartpole | |
| --- | --- | --- | --- | --- | --- | --- | --- |
| | **Accuracy** | **Accuracy** | **Loss** | **MAE val** | **MAE test** | **nAUC** | **nFSD** |
| AdamW+NL1 | **0.9422** | 0.9800 | 0.0572 | 0.1555 | 0.1215 | **0.8449** | 0.155 |
| AdamW+ | 0.9398 | 0.9807 | 0.0551 | 0.1603 | 0.1338 | 0.8367 | 0.200 |
| AdamW | 0.9391 | 0.9804 | 0.0616 | 0.1690 | 0.1312 | 0.8225 | 0.205 |
| AMSGrad+ | 0.9388 | 0.9803 | 0.0638 | 0.1663 | 0.1349 | 0.8165 | 0.230 |
| AMSGrad | 0.9406 | 0.9801 | 0.0636 | 0.1684 | 0.1336 | 0.8083 | 0.245 |
| LAMB+ | 0.9364 | **0.9828** | **0.0443** | **0.1230** | **0.0913** | 0.6836 | 0.005 |
| LAMB | 0.9352 | 0.9825 | 0.0463 | **0.1275** | **0.0973** | 0.6336 | 0.000 |
| Adam+ (SNR lr) | - | - | - | - | - | **0.8579** | **0.420** |
| Adam+ | 0.9327 | 0.9815 | 0.0547 | 0.1621 | 0.1332 | **0.8577** | 0.330 |
| Adam | 0.9323 | 0.9811 | 0.0610 | 0.1681 | 0.1329 | 0.8149 | 0.120 |
| ADOPT | 0.9388 | 0.9813 | 0.0548 | 0.1581 | 0.1280 | 0.8263 | 0.160 |
| ADOPT+ | 0.9374 | 0.9814 | 0.0560 | 0.1592 | 0.1339 | 0.8190 | 0.280 |
| AdaBound | **0.9428** | 0.9821 | 0.0882 | 0.1707 | 0.1460 | 0.0322 | 0.000 |
| AdaBeliefW | 0.9402 | 0.9804 | 0.0597 | 0.1639 | 0.1262 | 0.8334 | 0.145 |
| Lion ($\beta_2 = 0.99$) | 0.9372 | 0.9773 | **0.0335** | **0.1090** | **0.0863** | 0.4564 | 0.000 |
| AdaBelief | 0.9326 | 0.9804 | 0.0549 | 0.1583 | 0.1271 | 0.7984 | 0.120 |
| PIDAOSI* (Chen et al., 2024) | **0.9452** | **0.9836** | **0.0441** | 0.1439 | 0.1100 | 0.8344 | **0.354** |
| RMSprop ($\beta_1 = 0$) (Tieleman & Hinton, 2012) | 0.9281 | **0.9843** | 0.0597 | 0.1698 | 0.1459 | 0.7899 | 0.146 |
| AdaShift (Zhou et al., 2019) | 0.9346 | 0.9775 | 0.1687 | 0.2286 | 0.1958 | 0.3760 | 0.000 |

∗ Training followed the default PIDAOSI parameters for MNIST and CIFAR-10 as in Table 6 of (Chen et al., 2024); for Cartpole, the parameters of PIDAOSI are set to identical to that of the CIFAR-10 test.

Table 11: Training hyperparameters for ZINC dataset with GPS graph transformer

| Hyperparameter | Value |
| --- | --- |
| Dataset | ZINC (Gómez-Bombarelli et al., 2018) |
| Model | 10-layer GPS graph transformer (Rampášek et al., 2022) |
| Channels | 64 |
| Positional Encoding (PE) | Random Walk, with length 20 |
| PE dim | 8 |
| Attention type | Multihead |
| Dropout | 0.5 on attention head |
| Batch size | 128 |
| Epochs | 1000 |
| Initial Learning rate (lr) | $1 \times 10^{-3}$ for LAMB and LAMB+; $1 \times 10^{-4}$ for the rest |
| Learning rate scheduler | Constant (Table 12); Cosine annealing (Table 13) |
| Weight decay | $1 \times 10^{-5}$ |
| Loss function | Mean Absolute Error (MAE) |
| Random seed | 51 (constant lr in Table 12), 41 (cosine lr in Table 13) |

enhanced version generally achieve better performance for smaller $\beta$s, implying a more accurate estimation of noise power with out enhancement.

Next, we evaluate the performance of GT under a cosine annealing learning rate schedule, which are reported in the Figs. 9 and the Table 13. In this setting, Lion no longer outperforms all other methods, in fact, its lowest training loss did not translate to best validation and test MAE. This shows that Lion is highly sensitive to hyperparameters, in our other tests with a different random seed, Lion also spikes in training loss and does not recovery well afterwards. In contrast, Adam+NL3

Table 12: Molecular graph regression with graph transformers under fixed learning rate (**smaller is better**). Highlights are the **best** within each group, and the top first, second, and third.

| $(\beta_1, \beta_2)$ | Optimizer (lr) | Loss | MAE val | MAE test | Optimizer (lr) | Loss | MAE val | MAE test |
|---|---|---|---|---|---|---|---|---|
| 0.5, 0.5 | AdamW+NL3 | 0.0459 | 0.1339 | 0.1017 | Adam+NL3 | 0.0483 | **0.1249** | 0.1019 |
| 0.9, 0.9 | | 0.0418 | 0.1304 | 0.0944 | | 0.0469 | 0.1276 | 0.1001 |
| 0.9, 0.999 | | 0.0425 | 0.1249 | 0.0983 | | 0.0431 | 0.1279 | 0.0999 |
| 0.5, 0.5 | AdamW+NL2 | 0.0429 | 0.1253 | 0.1008 | Adam+NL2 | 0.0466 | 0.1295 | 0.0981 |
| 0.9, 0.9 | | 0.0433 | 0.1261 | 0.0945 | | **0.0463** | 0.1270 | **0.0970** |
| 0.9, 0.999 | | 0.0435 | 0.1299 | 0.0966 | | 0.0480 | 0.1318 | 0.1017 |
| 0.5, 0.5 | AdamW+NL1 | 0.0459 | 0.1422 | 0.1111 | Adam+NL1 | 0.0472 | 0.1412 | 0.1091 |
| 0.9, 0.9 | | 0.0533 | 0.1569 | 0.1240 | | 0.0514 | 0.1559 | 0.1268 |
| 0.9, 0.999 | | 0.0572 | 0.1555 | 0.1215 | | 0.0581 | 0.1613 | 0.1252 |
| 0.5, 0.5 | AdamW+ | 0.0532 | 0.1520 | 0.1182 | Adam+ | 0.0530 | 0.1534 | 0.1196 |
| 0.9, 0.9 | | 0.0548 | 0.1594 | 0.1273 | | 0.0599 | 0.1645 | 0.1302 |
| 0.9, 0.999 | | 0.0551 | 0.1603 | 0.1338 | | 0.0547 | 0.1621 | 0.1332 |
| 0.5, 0.5 | AdamW | 0.0667 | 0.1662 | 0.1315 | Adam | 0.0673 | 0.1697 | 0.1299 |
| 0.9, 0.9 | | 0.0620 | 0.1706 | 0.1307 | | 0.0620 | 0.1650 | 0.1297 |
| 0.9, 0.999 | | 0.0616 | 0.1690 | 0.1312 | | 0.0610 | 0.1681 | 0.1329 |
| 0.5, 0.5 | LAMB+ | 0.0560 | 0.1479 | 0.1225 | ADOPT+ | **0.0547** | **0.1456** | **0.1174** |
| 0.9, 0.9 | | 0.0466 | 0.1266 | 0.0955 | | 0.0608 | 0.1588 | 0.1292 |
| 0.9, 0.999 | | **0.0443** | 0.1230 | 0.0913 | | 0.0548 | 0.1582 | 0.1281 |
| 0.5, 0.5 | LAMB | 0.0588 | 0.1485 | 0.1207 | ADOPT | 0.0583 | 0.1568 | 0.1297 |
| 0.9, 0.9 | | 0.0467 | 0.1268 | 0.0952 | | 0.0536 | 0.1579 | 0.1281 |
| 0.9, 0.999 | | 0.0463 | 0.1276 | 0.0974 | | 0.0560 | 0.1593 | 0.1339 |
| 0.5, 0.5 | AMSGrad+ | 0.0938 | 0.1989 | 0.1602 | AMSGrad | 0.1268 | 0.2157 | 0.1853 |
| 0.9, 0.9 | | 0.0850 | 0.1867 | 0.1587 | | 0.0879 | 0.1885 | 0.1542 |
| 0.9, 0.999 | | **0.0638** | **0.1663** | **0.1349** | | 0.1718 | 0.2434 | 0.2098 |
| 0.9, 0.999 | AdaBelief | 0.0549 | 0.1583 | 0.1271 | RMSprop ($\beta_1=0$) | 0.0597 | 0.1698 | 0.1459 |
| 0.9, 0.999 | AdaShift | 0.1687 | 0.2286 | 0.1958 | AdaBound | 0.0882 | 0.1707 | 0.1460 |
| Custom | Lion | 0.0335 | 0.1090 | 0.0863 | PIDAOSI | 0.0441 | 0.1439 | 0.1100 |

and AdamW+NL3 yield an important result: although initially slower, as can be seen in the Fig. 9, they eventually attain the lowest validation and test MAE. This improvement can be attributed to their retention of gradient magnitude, which enables more confident update steps, particularly under diminishing learning rates. The results in Table 13 also confirm that our enhanced optimizers consistently outperform the corresponding baselines within most of the groups, except for LAMB.

This experiment suggests the possibility of using smaller smoothing factors ($\beta$s) for our enhanced Adam variants in training transformer architecture, which would be impossible without the replacement of second raw moment in Adam variants with second central moment.

Table 13: Molecular graph regression with graph transformers under cosine learning rate annealing (**smaller is better**). Highlights are the **best** within each group, and the top first, second, and third.

| $(\beta_1, \beta_2)$ | Optimizer | | MAE | | Optimizer | | MAE | |
|---|---|---|---|---|---|---|---|---|
| | | Loss | val | test | | Loss | val | test |
| 0.5, 0.5 | | 0.0268 | 0.1326 | 0.1051 | | 0.0255 | 0.1313 | 0.1019 |
| 0.9, 0.9 | AdamW+NL3 | **0.0164** | **0.1142** | **0.0859** | Adam+NL3 | **0.0166** | 0.1180 | **0.0834** |
| 0.9, 0.999 | | 0.0175 | 0.1168 | 0.0912 | | 0.0167 | 0.1161 | 0.0922 |
| 0.5, 0.5 | | 0.0351 | 0.1440 | 0.1108 | | 0.0342 | 0.1437 | 0.1069 |
| 0.9, 0.9 | AdamW+NL2 | 0.0184 | 0.1258 | 0.0899 | Adam+NL2 | 0.0183 | 0.1218 | 0.0912 |
| 0.9, 0.999 | | 0.0190 | 0.1219 | 0.1015 | | 0.0186 | 0.1209 | 0.0970 |
| 0.5, 0.5 | | 0.0429 | 0.1548 | 0.1170 | | 0.0425 | 0.1527 | 0.1213 |
| 0.9, 0.9 | AdamW+NL1 | 0.0396 | 0.1499 | 0.1192 | Adam+NL1 | 0.0456 | 0.1586 | 0.1182 |
| 0.9, 0.999 | | 0.0504 | 0.1568 | 0.1226 | | 0.0503 | 0.1588 | 0.1295 |
| 0.5, 0.5 | | 0.0365 | 0.1509 | 0.1172 | | 0.0326 | 0.1444 | 0.1139 |
| 0.9, 0.9 | AdamW+ | 0.0523 | 0.1596 | 0.1243 | Adam+ | 0.0534 | 0.1666 | 0.1265 |
| 0.9, 0.999 | | 0.0579 | 0.1657 | 0.1276 | | 0.0596 | 0.1678 | 0.1301 |
| 0.9, 0.999 | AdamW | 0.0543 | 0.1711 | 0.1316 | Adam | 0.0546 | 0.1703 | 0.1318 |
| 0.9, 0.999 | LAMB+ | 0.0179 | 0.1208 | 0.0914 | LAMB | 0.0179 | **0.1183** | **0.0891** |
| 0.5, 0.5 | | **0.0384** | **0.1476** | **0.1131** | | 0.0484 | 0.1610 | 0.1256 |
| 0.9, 0.9 | ADOPT+ | 0.0534 | 0.1633 | 0.1297 | ADOPT | 0.0544 | 0.1654 | 0.1301 |
| 0.9, 0.999 | | 0.0567 | 0.1651 | 0.1354 | | 0.0582 | 0.1656 | 0.1329 |
| 0.9, 0.999 | AMSGrad+ | **0.0646** | **0.1656** | **0.1346** | AMSGrad | 0.0648 | 0.1733 | 0.1399 |
| 0.9, 0.999 | AdaBelief | 0.0559 | 0.1676 | 0.1313 | RMSprop ($\beta_1 = 0$) | 0.0553 | 0.1564 | 0.1286 |
| 0.9, 0.999 | AdaShift | 0.1917 | 0.2435 | 0.2107 | AdaBound | 0.0402 | 0.1377 | 0.1157 |
| Custom | Lion | **0.0140** | **0.1217** | **0.0891** | PIDAOSI | 0.0590 | 0.1598 | 0.1236 |

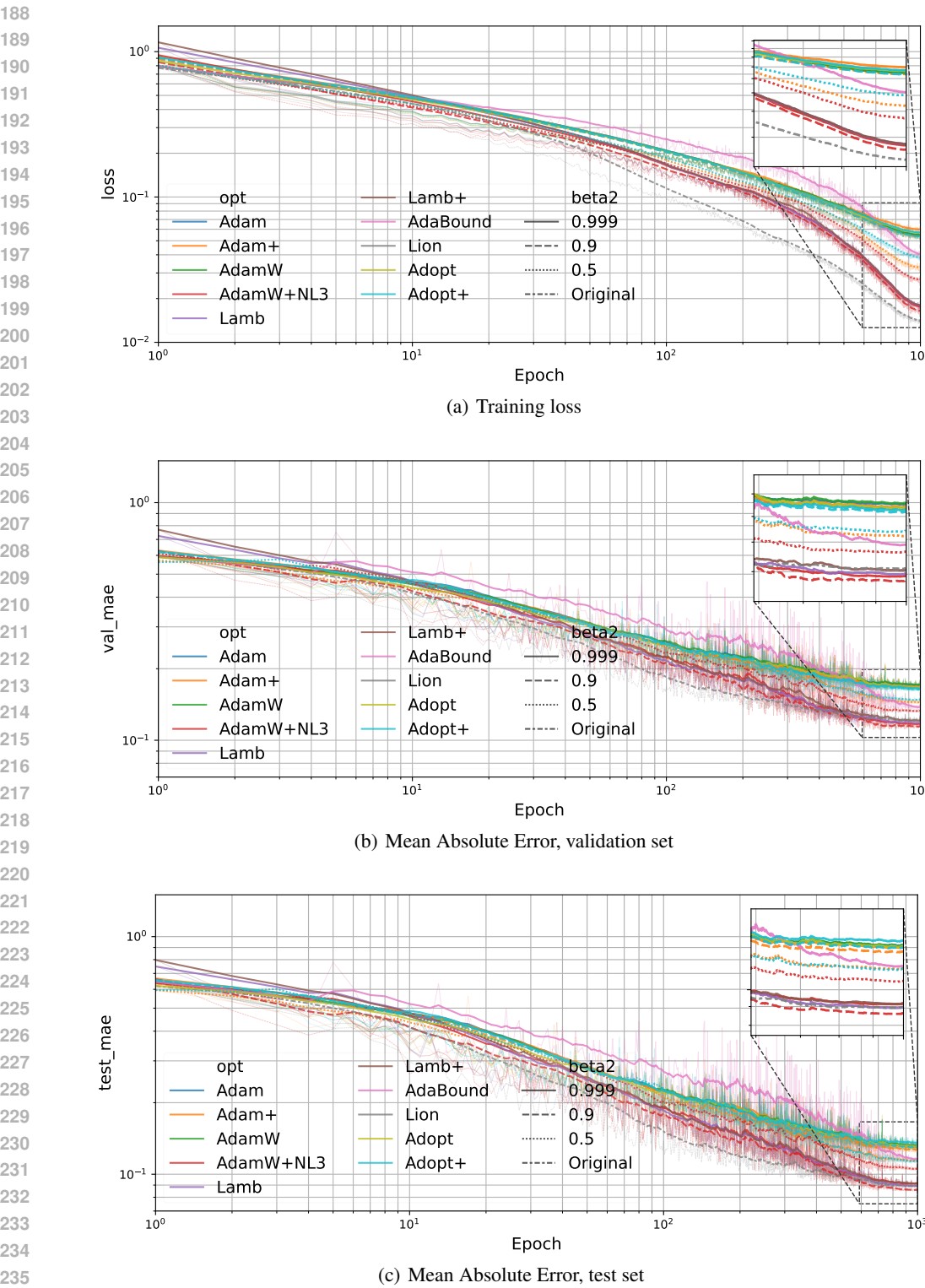

(a) Training loss

(b) Mean Absolute Error, validation set

(c) Mean Absolute Error, test set

Figure 9: Molecular graph property regression with GPS graph transformer on ZINC dataset, with cosine learning rate annealing: (a) Training loss MAE, (b) validation MAE, (c) test MAE. Notice that the validation set has not been used for learning rate scheduling, hence is just another a test set.

## G  HYPERPARAMETERS AND TEST RESULTS FOR CARTPOLE-DQN

To illustrate the benefits of our enhanced Adam optimizers in reinforcement learning, we first evaluate discrete control with DQN in the CartPole environment, with the hyperparameters listed in Table 14. The training return over global steps of five selected optimizers are presented in Fig. 10 and the full results of various optimizers and their parameter settings are listed in Table 10.

Table 14: Training hyperparameters for CartPole-v1 with DQN

| Hyperparameter | Value |
| --- | --- |
| Environment | CartPole-v1 |
| Algorithm | Deep Q-Network (DQN) (Mnih et al., 2015) |
| Code base | *dqn.py* in CleanRL (Huang et al., 2022) |
| Random seeds | $\{20, 21, 22, 23, 24, 25, 26, 27, 28, 29\}$ |
| Replay buffer size | 5000 |
| Batch size | 128 |
| Gamma (discount factor) | 0.99 |
| Learning rate | $2.5 \times 10^{-4}$ |
| Train start | After 10,000 steps |
| Train freq | Every 10 steps |
| Target network update freq | Every 500 steps |
| Exploration schedule | $\epsilon$-greedy (linear decay from 1.0 to 0.05) |
| Exploration fraction | 0.1 (50,000) steps |
| End exploration rate | 0.05 |
| Max global steps | 500,000 |
| Weight decay | $1 \times 10^{-5}$ |
| Loss function | MSE |

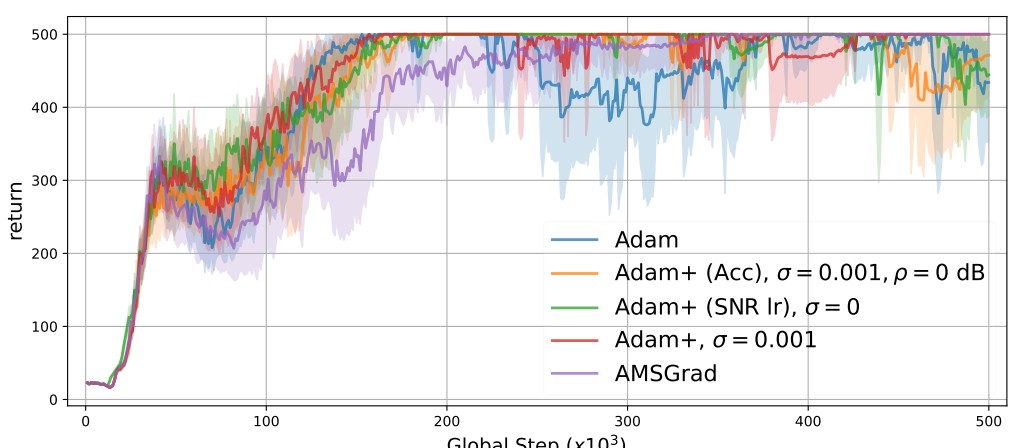

Figure 10: DQN with Cartpole-v1, average over 10 seeds (20-29), 95% confidence interval in errorband

To evaluate the performance of different optimizers for training DQN in CartPole, we run a total of 10 training trajectories under 10 different random seeds (20-29), and present the average return under the 10 seeds over the number of environment interactions (global steps). This representation is fair since global steps represents the actual computing time for training compared to the number of episodes, since an episode may contain different numbers of steps due to early termination.

To accurately reflect the performance of DQN, we split the total 500,000 global steps into chunks of 1000 steps, and find the average return of each chunk for each seed. In Fig. 10, the average return and its 95% confidence interval based on 10 different seeds are presented. In this way, we place equal weights on different seeds in the averaging. This approach avoid over-representing failing cases that would record small returns more frequently due to early termination of each episode.

Table 15: Performance of Optimizers (**larger is better**) for Cartpole DQN at $\beta_1 = 0.9, \beta_2 = 0.999$, from the average of 10 runs (random seeds 20-29) and reported using two metrics: normalized area under the curve (nAUC) and the normalized full score duration (nFSD). Highlights are the top **first**, **second**, and **third**.

| Optimizer | Hyperparameters | nAUC | nFSD |
|---|---|---|---|
| Adam+ (SNR lr) | $\sigma = 0$ | **0.8580** | **0.420** |
| Adam+ | $\sigma = 0.001$ | **0.8577** | **0.330** |
| AdamW+NL1 | $\sigma = 0$ | **0.8450** | 0.155 |
| AdamW+ | $\sigma = 0.001$ | 0.8367 | 0.200 |
| Adam+NL1 | $\sigma = 0$ | 0.8355 | 0.120 |
| PIDAOSI (Chen et al., 2024) | default | 0.8344 | **0.354** |
| AdaBeliefW | | 0.8335 | 0.145 |
| ADOPT | | 0.8264 | 0.160 |
| AdamW | | 0.8225 | 0.205 |
| ADOPT+ | $\sigma = 0$ | 0.8190 | 0.280 |
| AMSGrad+ | $\sigma = 0$ | 0.8166 | 0.230 |
| Adam | | 0.8150 | 0.120 |
| AMSGrad | | 0.8083 | 0.245 |
| AdaBelief | | 0.7985 | 0.120 |
| RMSProp (Tieleman & Hinton, 2012) | | 0.7899 | 0.146 |
| Adam+NL2 | $\sigma = 0$ | 0.7858 | 0.020 |
| Adam+NL3 | $\sigma = 0$ | 0.7585 | 0.010 |
| LAMB+ | $\sigma = 0$ | 0.6837 | 0.005 |
| LAMB | | 0.6337 | 0.000 |
| Lion ($\beta_2 = 0.99$) | default | 0.4564 | 0.000 |
| AdaShift (Zhou et al., 2019) | | 0.3760 | 0.000 |
| AdaBound | | 0.0323 | 0.000 |
| AdaGrad | | 0.0213 | 0.000 |

The five evaluated optimizers in Fig. 10 reach their initial convergence (reaching full return) during the global step window between 155k to 200k global steps, however, due to disruptions from exploration (0.05 end exploration rate), most of them experience drops in return after the initial convergence. This instability can be attributed to a relatively high final exploration ratio of 0.05, which, while promoting generalization, introduces increased variance in performance. Visually, it is evident that our enhanced variants achieve faster convergence than AMSGrad and exhibit greater stability compared to Adam.

More specifically, the noise injection introduced in Adam+, $\sigma = 0.001$ allows for faster convergence and higher returns compared to Adam. Adam+ (SNR lr), $\sigma = 0$, shows that adapting the learning rate based on SNR results in smooth and consistent performance, showing one of the most stable trajectories.

To further quantify the visual inspection, we consider two performance metrics, a normalized Area Under Curve (nAUC), and normalized full score duration (nFSD), which measure the portion of chunks with full score return. These quantitative results are partially listed in Table 10 and comprehensively in Table 15 which contains more optimizers and hyperparameter combinations for our enhanced optimizers. Adam+ (SNR lr), $\sigma = 0$ achieves the highest nAUC and nFSD, demonstrating the benefit of SNR-adaptive learning rate. The Adam+ with noise injection, $\sigma = 0.001$, achieves a very competitive performance and is ranked second in nAUC.

On the other hand, signed and logarithmic optimizers perform poorly, showing that reinforcement learning depends on linear (or quadratic) optimizers for fast response. AdamW and LAMB also perform poorly due to the small and shallow neural network architecture in this task.

## H CONVERGENCE ANALYSIS ON A QUADRATIC FUNCTION

We study the behavior of Adam+ and AdaBelief on the one-dimensional stochastic quadratic model

$$\min_{\theta \in \mathbb{R}} J(\theta) = \frac{\lambda}{2}(\theta - \theta^\star)^2, \qquad \lambda > 0, \tag{13}$$

with stochastic gradients

$$g_t = \lambda(\theta_t - \theta^\star) + \xi_t, \qquad \xi_t \sim \mathcal{N}(0, \sigma^2) \text{ i.i.d.} \tag{14}$$

We analyze the asymptotic behavior of the optimizer near the optimum $\theta^\star$. In this regime, the gradient is dominated by the stochastic noise, i.e., $g_t \approx \xi_t$ where $\xi_t \sim \mathcal{N}(0, \sigma^2)$ (Balles & Hennig, 2018). Since the input to the second-moment estimator is effectively i.i.d., the estimator $v_t$ converges to a stationary distribution.

Furthermore, to isolate the effect of $v_t$ (the distinct feature of Adam+ and AdaBelief), we abstract away the effect of the first-order moment. We approximate $m_t \approx g_t$, treating the numerator as an unfiltered gradient. Consequently, both optimizers reduce to the simplified update rule

$$\theta_{t+1} = \theta_t - \alpha \frac{g_t}{\sqrt{v_t}}. \tag{15}$$

We omit the bias-correction terms as is often done in the related work Zhuang et al. (2020).

Define the error $x_t = \theta_t - \theta^\star$. With that (15) becomes

$$x_{t+1} = \left(1 - \frac{\lambda\alpha}{\sqrt{v_t}}\right) x_t - \alpha \frac{\xi_t}{\sqrt{v_t}}. \tag{16}$$

To derive the convergence rate, we employ a *mean-field approximation*, replacing the random variable $v_t$ with its steady-state expectation $\mathbb{E}[v_t]$. This approximation assumes that the fluctuations of $v_t$ around its mean are negligible compared to the dynamics of the parameter $\theta_t$, a standard simplification in the analysis of adaptive moment estimation (Balles & Hennig, 2018; Défossez et al., 2022).

Under this assumption, $v_t$ is treated as a constant scalar $\mathbb{E}[v_t]$, simplifying (16) to the stochastic linear recursion

$$x_{t+1} = \rho\, x_t - \eta\, \xi_t, \qquad \rho := 1 - \frac{\lambda\alpha}{\sqrt{\mathbb{E}[v_t]}}, \quad \eta := \frac{\alpha}{\sqrt{\mathbb{E}[v_t]}}. \tag{17}$$

Since $\xi_t$ has zero mean and is independent of $x_t$, the mean error satisfies

$$\mathbb{E}[x_{t+1}] = \rho\, \mathbb{E}[x_t], \tag{18}$$

so the mean converges whenever $|\rho| < 1$.

An optimizer will converge to the error ball of a radius given by the variance of the error $x_t$. This yields

$$\text{Var}[x_{t+1}] = \rho^2 \text{Var}[x_t] + \eta^2 \text{Var}[\xi_t] = \rho^2 \text{Var}[x_t] + \frac{\alpha^2}{\mathbb{E}[v_t]}\sigma^2, \tag{19}$$

due to the mean-field approximation of $v_t$, and the i.i.d. assumption of the gradient noise.

For $|\rho| < 1$, the steady-state variance is given by

$$\text{Var}[x_\infty] := \lim_{t \to \infty} \text{Var}[x_t] = \frac{\alpha^2 \sigma^2}{\mathbb{E}[v_t](1 - \rho^2)}. \tag{20}$$

All optimizer-dependent behavior now arises from the stationary scale $\mathbb{E}[v_t]$.

The second central moment $v_t$ is constructed from the centered quantity $g_t - w_t$, where $w_t$ is an exponential moving average with parameter $\beta$, such that $\beta = \beta_1$ for AdaBelief and $\beta = \beta_2$ for Adam+

$$w_t = \beta w_{t-1} + (1 - \beta)g_t \tag{21}$$

$$v_t = \beta v_{t-1} + (1 - \beta)(w_t - g_t)^2. \tag{22}$$

Then the mean-field approximation equals

$$\mathbb{E}[v_t] = \mathbb{E}\left[\beta v_{t-1} + (1 - \beta)(g_t - w_t)^2\right] \tag{23}$$

$$= \beta\mathbb{E}[v_{t-1}] + \mathbb{E}\left[(g_t - w_t)^2\right] - \beta\mathbb{E}\left[(g_t - w_t)^2\right] \tag{24}$$

At the steady-state, $\mathbb{E}[v_{t-1}] = \mathbb{E}[v_t]$. Then based on Lemma H.1 we obtain

$$\mathbb{E}[v_t] = \mathbb{E}\left[(g_t - w_t)^2\right] = \frac{2\beta^2}{1+\beta}\sigma^2. \tag{25}$$

Substituting (25) into (20) gives

$$\mathrm{Var}[x_\infty] = \frac{\alpha^2}{1-\rho^2}\frac{1+\beta}{2\beta^2} \tag{26}$$

$$\rho = 1 - \frac{\lambda\alpha\sqrt{1+\beta}}{\sqrt{2}\beta\sigma}. \tag{27}$$

By plugging (27) into (26) and assuming sufficiently small $\alpha$ such that we can ignore second-order terms arising in $\rho^2$ we get

$$\mathrm{Var}[x_\infty] = \frac{\alpha^2}{1-\rho^2}\frac{1+\beta}{2\beta^2} \approx \frac{\alpha\sigma\sqrt{1+\beta}}{2\sqrt{2}\lambda\beta}. \tag{28}$$

This expression quantifies the trade-off between AdaBelief ($\beta = \beta_1$) and Adam+ ($\beta = \beta_2$): for the default choice $\beta_1 < \beta_2$,

- the contraction factor $|\rho|$ is smaller for AdaBelief, giving *faster convergence*
- the variance of the steady-state error is a monotonically decreasing function in $\beta$. It is smaller for Adam+, giving *smaller steady-state error*.

This captures the fundamental trade-off: AdaBelief converges faster but to a noisier solution, while Adam+ converges more slowly, but to a more accurate one due to a more faithful estimation of the gradient signal-to-noise ratio.

**Lemma H.1** (Steady-state variance estimate). *Under the model* (14) *and the EMA definition* (21)*, and assuming stationarity of $w_t$, we have*

$$\mathbb{E}[v_t] = \mathbb{E}\left[(g_t - w_t)^2\right] = \frac{2\beta^2}{1+\beta}\sigma^2. \tag{29}$$

*Proof.* From (14) we can write
$$g_t = \bar{g} + \xi_t.$$
Taking expectations in (21) and using linearity,
$$\mathbb{E}[w_t] = \beta\mathbb{E}[w_{t-1}] + (1-\beta)\mathbb{E}[g_t].$$
In steady state, $\mathbb{E}[w_t] = \mathbb{E}[w_{t-1}] =: \bar{w}$, and $\mathbb{E}[g_t] = \bar{g}$, so
$$\bar{w} = \beta\bar{w} + (1-\beta)\bar{g} \quad \Rightarrow \quad \bar{w} = \bar{g}.$$
Thus, we can decompose
$$w_t = \bar{g} + z_t,$$
where $z_t$ is a zero-mean random process capturing the EMA of the noise. In particular,
$$g_t - w_t = (\bar{g} + \xi_t) - (\bar{g} + z_t) = \xi_t - z_t. \tag{30}$$
Then using the decomposition in (21),
$$w_t = \beta w_{t-1} + (1-\beta)\bar{g} + (1-\beta)\xi_t,$$
and hence
$$z_t = w_t - \bar{g} = \beta(w_{t-1} - \bar{g}) + (1-\beta)\xi_t = \beta z_{t-1} + (1-\beta)\xi_t.$$
Since $\xi_t$ is independent of $z_{t-1}$ and has zero mean, the variance of $z_t$ satisfies
$$\mathrm{Var}[z_t] = \beta^2\mathrm{Var}[z_{t-1}] + (1-\beta)^2\mathrm{Var}[\xi_t] = \beta^2\mathrm{Var}[z_{t-1}] + (1-\beta)^2\sigma^2.$$
At stationarity, $\mathrm{Var}[z_t] = \mathrm{Var}[z_{t-1}]$, giving
$$\mathrm{Var}[z_t] = \frac{1-\beta}{1+\beta}\sigma^2.$$

Since $w_t = \bar{g} + z_t$, we also have $\mathrm{Var}[w_t] = \mathrm{Var}[z_t]$.

From (21) we have

$$g_t - w_t = g_t - \big(\beta w_{t-1} + (1-\beta)g_t\big) = \beta(g_t - w_{t-1}),$$

and using (30) yields

$$\mathbb{E}\left[(g_t - w_t)^2\right] = \beta^2 \mathbb{E}\left[(g_t - w_{t-1})^2\right] = \beta^2 \,\mathbb{E}\left[(\xi_t - z_{t-1})^2\right] = \beta^2\big(\sigma^2 + \mathbb{E}\left[z_{t-1}^2\right]\big),$$

using independence of $\xi_t$ and $z_{t-1}$. At stationarity $\mathbb{E}\left[z_{t-1}^2\right] = \mathbb{E}\left[z_t^2\right] = \mathrm{Var}[z_t]$, so

$$\mathbb{E}\left[(g_t - w_t)^2\right] = \beta^2\big(\sigma^2 + \mathrm{Var}[z_t]\big) = \frac{2\beta^2}{1+\beta}\,\sigma^2,$$

which is the desired expression. $\qquad\square$

**Comparison with standard Adam.** Finally, we compare the convergence behavior with standard Adam. Unlike Adam+ and AdaBelief, standard Adam uses an uncentered second moment estimator $v_t$. Under the mean-field approximation in the quadratic regime, the expected second moment for Adam is

$$\mathbb{E}\left[v_t^{\mathrm{Adam}}\right] \approx \mathbb{E}\left[g_t^2\right] = \mathrm{Var}[g_t] + (\mathbb{E}\left[g_t\right])^2 = \sigma^2 + \lambda^2 \mathbb{E}\left[x_t\right]^2. \tag{31}$$

Comparing this to the Adam+ estimator ($\mathbb{E}\left[v_t\right] \approx \sigma^2$), we observe that

$$\mathbb{E}\left[v_t^{\mathrm{Adam}}\right] > \mathbb{E}\left[v_t\right] \quad \text{whenever } \mathbb{E}\left[x_t\right] \neq 0. \tag{32}$$

The spectral radius $\rho$ (27) that governs the convergence is given by

$$\rho = 1 - \lambda\alpha\frac{1}{\sqrt{\mathbb{E}\left[v_t\right]}}. \tag{33}$$

Since the denominator for Adam is larger, the effective step size is smaller, resulting in a spectral radius closer to 1 (slower convergence). This result indicates that standard Adam dampens the useful gradient signal that unnecessarily slowing down the optimization process. Based on SNR-centering, Adam+ maintains a lower spectral radius that is a function only of the noise level.

To further support our derivations, we evaluate the convergence behavior of Adam+ and AdaBelief on the Rosenbrock function. The result is shown in the Fig. 11. While AdaBelief exhibits faster initial descent, Adam+ demonstrates lower final loss. Specifically, Adam+ converges closer to the global optimum and maintains a smaller oscillation radius (steady-state variance) around the optimal point compared to AdaBelief. This confirms our theoretical result that the additional state $w_t$ enables convergence to a smaller steady-state error ball.

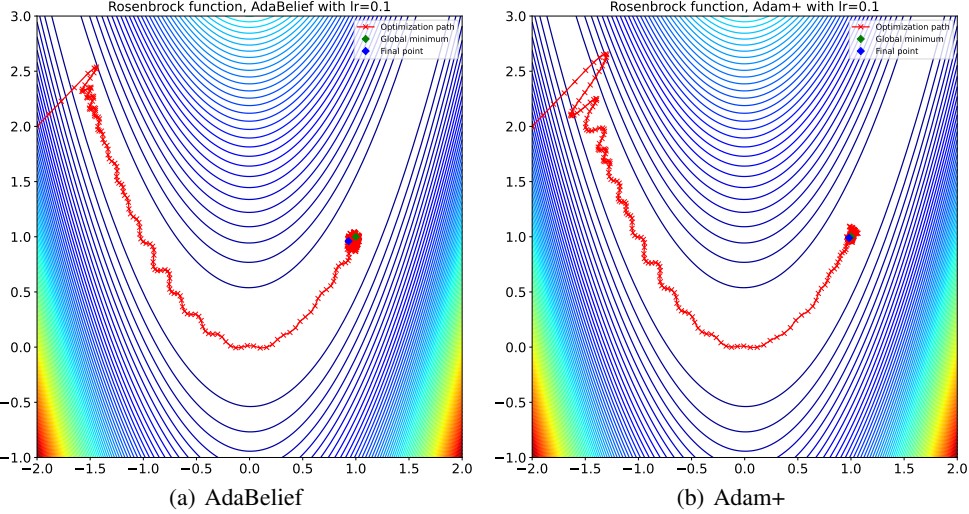

(a) AdaBelief $\qquad\qquad\qquad\qquad\qquad$ (b) Adam+

Figure 11: Convergence on the Rosenbrock function. Although AdaBelief achieves faster early-stage convergence, Adam+ ultimately reaches a lower loss value and settles into a tighter error bound around the optimum. This aligns with our theoretical analysis.

# I  HYPERPARAMETERS SAC IN MUJOCO ENVIRONMENTS

The hyperparameters and the results for SAC in the MuJoCo environment are given in Table 16.

Table 16: Training hyperparameters for SAC in MuJoCo environments

| Hyperparameter | Value |
| --- | --- |
| Environment | Hopper-v5, HalfCheetah-v5, Humanoid-v5 |
| Algorithm | Soft Actor-Critic (SAC) (Haarnoja et al., 2018) (Spinning Up) |
| Code base | *sac_continuous_action.py* in CleanRL (Huang et al., 2022) |
| Random seed | 1 |
| num_envs | 1 |
| Replay buffer size | $10^6$ |
| Batch size | 256 |
| Gamma | 0.99 |
| Tau | 0.005 |
| Alpha | 0.2 |
| Autotune | True |
| Learning rate | $3 \times 10^{-4}$ (for both policy and target networks) |
| Train start | After 10,000 steps |
| Policy network train freq | Every 1 step |
| Target network update freq | Every 1 step |
| Max global steps | $\{1 \times 10^6, 3 \times 10^6, 3 \times 10^6\}$ |
| Weight decay | 0 (none) |

# J  ABLATION: DISENTANGLING THE NOISE INJECTION EFFECT

To understand whether the noise injection effect is complementary or orthogonal to SNR-centering, we conducted an ablation in which we added the noise of $\sigma \approx 0.03$ ($-50$dB) and $\sigma = 0.01$ ($-60$dB) to both Adam and Adam+ when training an RL agent on the Hopper-v5 environment. The hyperparameters are given in the Table 16. The results averaged over 3 seeds are shown in Fig. 12. Three observations are in order: first, for the noiseless case, there is a clear effect of SNR-centering resulting in a higher return of Adam+ compared to vanilla Adam. Second, the noise injection of $\sigma \approx 0.03$ improves the return of Adam+ toward the end of the training. For the same $\sigma$, Adam also exhibits a performance boost at the final training steps. For a lower noise level of $\sigma = 0.01$, both optimizers experience a performance drop. This indicates that for the considered setting, the noise injection effect is orthogonal to SNR-centering, i.e., Adam can also benefit from it, and it should be treated as a hyperparameter that encourages exploration, achieving an effect similar to that of $\epsilon$-greedy.

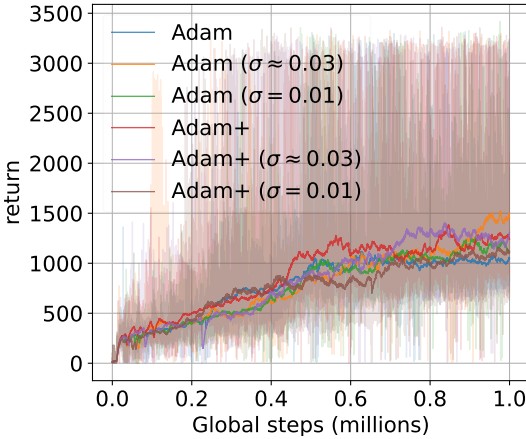

Figure 12: Noise injection effect on Adam and Adam+ optimizers for different levels of noise $\sigma$ in Hopper-v5 environment.

### J.1 Use of Large Language Models (LLMs)

We used LLMs as a writing assistant during the preparation of this manuscript. Specifically, it was employed to (i) improve the clarity and fluency of text passages, (ii) suggest alternative phrasings, and (iii) help structure certain sections.

