# OpenReview forum: "Boosting Adam-like Optimizers with Signal-to-Noise Ratio Guided Updates"
_ICLR.cc/2026/Conference — Submitted to ICLR 2026_

### Official Review · Reviewer_KLCS · 2025-10-25

**Soundness:** 2
**Presentation:** 2
**Contribution:** 2
**Rating:** 2
**Confidence:** 5

**Summary:**

The paper proposes **Adam+**, an enhancement of the Adam-like optimizers that replaces the second raw moment with the *variance* to better capture gradient noise via the signal-to-noise ratio (SNR). This centering yields more accurate normalization of updates and improves stability. Adam+ also adds mild noise injection to reduce bias between moment estimates. Experiments across reinforcement learning, sequence modeling, and graph regression show faster convergence, higher returns, and greater robustness than Adam and its variants, suggesting that centering second moments is a simple yet broadly effective improvement for adaptive optimizers.

**Strengths:**

1. The experiments span a diverse range of tasks, including reinforcement learning (MuJoCo, Atari), sequence modeling (LSTM, nanoGPT, BERT), and graph regression, supported by clear plots that demonstrate consistent performance gains of Adam+ over baselines.
2. The proposed SNR-guided update is conceptually simple and easy to implement, providing an intuitive link between gradient statistics and learning dynamics while requiring only minimal modification to existing Adam-like optimizers.

**Weaknesses:**

1. The paper lacks theoretical justification for why signal-to-noise ratio–guided updates should improve optimization. It does not provide convergence analysis or formal guarantees showing whether Adam+ achieves faster or more stable convergence than Adam.

2. Adam+ introduces an additional memory cost proportional to model parameters to store the momentum term $w_t$. The authors should explicitly discuss this overhead and justify its practicality, especially for memory-constrained scenarios such as large-scale language model training.

3. Figure 6 appears overly dense and visually cluttered. The authors should consider improving its layout or simplifying the presentation to enhance readability and interpretability.

4. The experimental scale remains limited, focusing mainly on small or toy settings. Larger-scale benchmarks would strengthen the empirical claims and demonstrate practical relevance for real-world training workloads.

5. The discussion of related work is insufficient and omits several recent optimizers. Moreover, the baselines used in experiments are relatively weak considering current advances in optimization methods. In particular, comparisons with AdaEMAMix [1], SOAP [2], Muon [3], and Scion [4] are missing and would provide a more comprehensive evaluation.

## References
[1] Pagliardini, Matteo, Pierre Ablin, and David Grangier. "The ademamix optimizer: Better, faster, older." arXiv preprint arXiv:2409.03137 (2024).
[2] Vyas, Nikhil, et al. "Soap: Improving and stabilizing shampoo using adam." arXiv preprint arXiv:2409.11321 (2024).
[3] Jordan, Keller et al. "Muon: An optimizer for hidden layers in neural networks." 2024. URL: https://kellerjordan.github.io/posts/muon/.
[4] Pethick, Thomas, et al. "Training deep learning models with norm-constrained lmos." arXiv preprint arXiv:2502.07529 (2025).

**Questions:**

See weaknesses above.

---

> ### Author Response · Authors · 2025-11-21
>
> We thank reviewer for review.
> > The paper lacks theoretical justification [...]
>
> We added the convergence analysis on a quadratic function in Appendix H. That one also includes Adam. Theoretically, standard Adam estimates the uncentered second moment $\mathbb{E}[g_t^2] = \sigma^2 + \|\nabla J\|^2$ from the gradient $g$ and the loss $J$. In the quadratic setting, this effectively increases the denominator: as the gradient signal $\|\nabla J\|$ increases, Adam reduces its effective step size. In contrast, Adam+ removes the signal component from the denominator ($\mathbb{E}[v_t] \approx \sigma^2$). Our analysis shows that this allows Adam+ to maintain a larger contraction rate during the convergence phase compared to Adam.
> > Adam+ introduces an additional memory cost [...]
>
> We acknowledge increase from $2|\theta|$ to $3|\theta|$ due to the estimator $w_t$, but justify it in the following way. First, this matches recent advances (e.g., the reviewer's suggested AdaEMAMix), which also use $\ge 3|\theta|$ states to separate fast/slow dynamics. This reflects a community consensus that trading memory for convergence stability is practical.
>
> Second, in RL, memory is dominated by the experience replay buffer (~$10^6$ transitions). The marginal cost of one parameter vector ($1|\theta|$) is negligible relative to the total system.
>
> Lastly regarding LLMs, modern infrastructure (ZeRO-Offload, FsDP) routinely handles $3|\theta|$ optimizers. Furthermore, the $w_t$ buffer is amenable to 8-bit storage, potentially lowering total memory below standard FP32 Adam.
>
> > Cluttered figure.
>
> Updated, thanks
>
> > The experimental scale remains limited [...]
>
> We'd like to clarify that the selected benchmarks, such as nanoGPT, Crammed BERT are not simply "toy" settings, but are recognized in the optimization community as proxies for large-scale pre-training dynamics.
>
> CrammedBERT is designed to simulate full-scale difficulties under a strict compute budget. Since the baseline hparams are already optimized for AdamW, our improvements indicate genuine algorithmic enhancement rather than poor baseline tuning.
>
> We agree that we did not train massive LLMs. This is because we prioritized breadth across diverse modalities (RL, graph regression, sequence modeling). The universality of gains across unrelated domains provides strong empirical evidence that SNR-centering addresses a fundamental optimization issue.
>
> Finally, we argue that demonstrating gains on benchmarks like Crammed BERT is highly relevant to "real-world training workloads". This is because the majority of practitioners work with finite computing budgets.
>
> > The discussion of related work is insufficient and omits several recent optimizers.[...]}
>
> Thanks for the comment.
> We expanded our Related Work section to explicitly discuss AdaEMAMix, SOAP, Muon, and Scion.
>
> Regarding the experimental baselines, we believe that our current selection is rigorous and appropriate for the specific domains we analyzed.
> 1. In our RL experiments, we compared against Adam with highly tuned hparams provided by the stable\_baselines3 "RL Zoo." In the RL community, the "RL Zoo" hparams represent a strong baseline that is difficult to beat.
> Relying on untuned or default configurations of recent optimizers (e.g., Muon or SOAP, which lack RL benchmarks) would likely constitute a weaker comparison.
> By beating the community-standard tuned Adam, we demonstrate that Adam+ provides genuine algorithmic gains that persist even against the best-known configurations of the baseline.
>
> 2. We emphasize that our evaluation did include a wide range of modern SOTA optimizers in our supervised learning tasks.
> In these settings, we compared Adam+ against Lion, ADOPT, AMSGrad, LAMB, AdaShift, and PIDAOSI. Adam+ remains competitive or superior against this extensive list of modern optimizers
>
> 3. The optimizers suggested by the reviewer (Muon, SOAP) are primarily designed for large-batch LLM pre-training. They often involve architectural assumptions (like eigen-preconditioning or tensor orthogonalization).
> Comparing Adam+ (a general-purpose variance correction) against these specialized LLM optimizers in an RL or graph regression context would introduce confounding variables, because they operate on fundamentally different algorithmic principles.
>
> Also, it is important to note that optimizers like Muon are often implemented as hybrid systems: they apply orthogonal updates to 2D tensors but rely on standard Adam/AdamW for 1D parameters (biases). Hence, Adam+ is not necessarily mutually exclusive to these methods. The SNR-based centering mechanism could potentially be integrated into the vector-update component of Muon to further enhance its performance.
>
> Therefore, our experimental design isolates the specific contribution of SNR-centering. By comparing Adam+ directly against its parent family (Adam/AdamW) across RL, graphs, and sequence modeling, we provide a controlled, "apples-to-apples" validation of the mechanism.

---

### Official Review · Reviewer_RYRY · 2025-10-27

**Soundness:** 2
**Presentation:** 2
**Contribution:** 1
**Rating:** 2
**Confidence:** 3

**Summary:**

This paper provides a variant of Adam called Adam+ by replacing the second raw moment with the second central moment (variance). The central idea is to view its gradient update as a momentum estimate normalized by noise amplitude. A comprehensive empirical result verifies the superiority of Adam+ over Adam in reinforcement learning and sequence modeling tasks.

**Strengths:**

- This paper provides detailed experimental results to illustrate the superiority of Adam+ over Adam or AdaBelief. The enhancement over training performance seems very clear from the figures and tables.

- The paper is well-written.

**Weaknesses:**

My major concerns mainly lie in the following.

- The motivation of Adam+ is not very clear. In particular, the essential differences between Adam+ and AdaBelief are not very clear. In my view, the only difference is to use $w_t$, which is also the exponential moving average of the gradient controlled by $\beta_2$, to replace $m_t$ in AdaBelief. I do not see the motivation for this replacement. The authors claim that ``While this formulation shares the use of a central
moment, its interpretation differs fundamentally from the SNR perspective presented in this work.", but without further explanation.

- The author also claims that the correlation between $m_t$ and the noise will hinder the convergence. First, I do not see any convincing experimental results to illustrate this. Perhaps some ablation studies are required. Second, with $m_t$ replaced by $w_t$, the correlation still exists in Adam+. Given this, I think that Adam+ may be just a slightly different version of AdaBelief.

- The author provides some experimental results to show that Adam+ performs better than Adam or AdaBelief. However, I think that the hyper-parameter setups are not very clear. The author only claims that ``Notice that we have used the same set of tuned
hyperparameters (Haarnoja et al., 2018; Raffin, 2020) for all simulations." without any further detailed explanation in the main body.

**Questions:**

- What are the major differences between Adam+ and AdaBelief?

- Which mechanisms lead to the performance enhancement of Adam+? Does this have some mathematical insights or intuition?

---

> ### Author Response · Authors · 2025-11-21
>
> > In particular, the essential differences between Adam+ and AdaBelief are not very clear. [...]}
>
> Please, see more detailed answer below and the convergence comparison in the Appendix H.
>
> > The author also claims that the correlation between $m_t$ and the noise will hinder the convergence [...]
>
> The proof that the correlation between $g_t$ (for $\beta_1=0)$ and $v_t$ (notice that this is not the *noise*, but the variance estimate) impairs Adam's convergence is given in Theorem 5 of AdaShift work (Zhou et. al., ICLR 2019). The argument why $\beta_1>0$ does not resolve the issue is given in the second paragraph in the subsection 3.3. We added a clearer reference to this Theorem.
>
> The reviewer is correct that Adam+, like AdaBelief, does not strictly break the temporal correlation between the numerator and denominator (unlike AdaShift), and we corrected the manuscript reflecting this change.
> However, Adam+ is fundamentally distinct from AdaBelief in what it estimates, and this distinction is critical for the steady-state performance.
>
> The difference lies in the timescale of the centering operation: AdaBelief (fast centering, $\beta_1$) centers the gradient using the fast first order moment $m_t$. This measures the "prediction error" of $m_t$, acting as a high-pass filter that reacts to local curvature changes. On the other hand, Adam+ (slow centering, $\beta_2$) centers the gradient using the slow mean $w_t$. This effectively mimics Welford’s online algorithm for variance estimation, aiming to capture the true gradient SNR relative to the stationary mean.
>
> While the temporal correlation remains in both, the bias of the estimator differs. Because of fast mean, AdaBelief treats the local changes in the loss curvature as the variance. On the other hand, Adam+ stabilizes around the true stationary variance. We have formalized this trade-off in Appendix H. Our theoretical analysis shows that under the standard setting ($\beta_1 < \beta_2$) AdaBelief has a smaller contraction factor $|\rho|$, leading to faster convergence. On the other hand Adam+ achieves a lower steady-state error variance. This confirms the intuition that while AdaBelief is more reactive (good for speed), Adam+ provides a more robust estimate of the gradient SNR (good for final performance).
>
> > The author provides some experimental results to show that Adam+ performs better than Adam or AdaBelief. However, I think that the hyper-parameter setups are not very clear. The author only claims that ``Notice that we have used the same set of tuned hyperparameters (Haarnoja et al., 2018; Raffin, 2020) for all simulations." without any further detailed explanation in the main body.
>
> To clarify the specific setups used, for robotic control and Atari (RL) we utilized the highly-tuned hyperparameter sets provided by the Stable-Baselines3 RL Zoo, as indicated in the paper. These represent the community standard for Adam/RMSProp performance in RL. By using these exact settings for Adam+, we demonstrate that our method improves performance without requiring the extensive re-tuning typically needed for new optimizers.
>
> For sequence modeling (Crammed BERT and LSTM), we adopted the specific optimized configurations reported by the original authors of Crammed  BERT and AdaBeleief for LSTM benchmark. Our goal was to perform a strict replacement test: we simply replaced the optimizer (AdamW and AdaBelief) while keeping the learning rate, batch size, and scheduler identical to the tuned baseline. The observed gains therefore reflect the algorithmic improvements rather than hyperparameter search advantages.
>
> For nanoGPT we utilized the default configuration from the official nanoGPT repository. Due to the computational cost of training large language models, we only verified if the improvement is visible under these standard hyperparameter settings.
>
> For molecular graph regression, we performed a finer evaluation. The complete grid of hyperparameters (including those under fixed learning rates and cosine annealing schedules) is documented in Table 12 and Table 13 of the Appendix.
>
> > What are the major differences between Adam+ and AdaBelief?
>
> AdaBelief vs. Adam+:
> 1. Fast vs. slow memory
> 2. Curvature vs. actual variance
> 3. Speed vs. accuracy
>
> Please see the reply above for more details.
>
> > Which mechanisms lead to the performance enhancement of Adam+? Does this have some mathematical insights or intuition?
>
> Adam+ mimics Welford's online variance estimator. More details in the reply above and in the theoretical analysis in the Appendix H.

---

> > ### Comment · Reviewer_RYRY · 2025-11-24
> >
> > I am still confused the comment that "Adam+ provides a more robust estimate of the gradient SNR (good for final performance)." Why Adam+, using a slow mean, could get a better accuracy. Are there any very initutive insights, such as theoretical results?

---

> > > ### Author Response · Authors · 2025-11-24
> > >
> > > Yes, in Appendix H, we analyze the convergence rate of Adam+ and AdaBelief. It turns out that AdaBelief converges faster, but Adam+ converges to the error ball of a smaller radius. The intuition is that near convergence, the fast mean of AdaBelief fluctuates with noise (because the EMA window is just $1/(1-0.9)=10$ samples long). Hence, the variance estimate contains a significant portion of high-frequency error.
> > >
> > > On the other hand, Adam+ uses a slow mean and an EMA window length of $\approx 1000$ samples. This provides a stationary reference point for the variance estimation, which, in turn, allows Adam+ to reach a smaller steady-state error.

---

### Official Review · Reviewer_PwsC · 2025-10-30

**Soundness:** 2
**Presentation:** 2
**Contribution:** 2
**Rating:** 4
**Confidence:** 5

**Summary:**

The paper reinterprets Adam through the lens of gradient signal-to-noise ratio, viewing the update as momentum scaled by an estimate of noise amplitude, and argues that Adam’s use of the raw second moment conflates mean and variance. The authors propose Adam+, which replaces the raw second moment with the central second moment by maintaining a slow EMA of gradients $w_t$ and updating $v_t$ with $(g_t - w_t)^2$. They also study a small Gaussian noise injection into gradients to decorrelate $m_t$ and the variance estimate and to avoid overly aggressive scaling when the variance becomes tiny. Across tasks in reinforcement learning, sequence modeling, molecular graph regression, and image classification, Adam+ and centered variants of other Adam-like optimizers show consistent gains or stability improvements over their baselines. In MuJoCo SAC and Atari DQN, Adam+ converges faster and is more robust to non-stationarity and exploration noise than Adam. In sequence modeling, Adam+ improves perplexities for LSTMs on Penn Treebank and reduces instability for nanoGPT relative to AdamW. GLUE fine-tuning with a crammed BERT shows average improvements with AdamW+ over AdamW. The paper supplements results with SNR diagnostics, ablations on the signed non-linear scaling of updates, and detailed hyperparameters to encourage reproducibility.

**Strengths:**

### Clear SNR-based formulation

The work offers a clean rethinking of Adam as scaling momentum by noise amplitude and shows that centering the second moment yields an estimator aligned with SNR. The resulting updates are simple, require only an extra EMA for the mean gradient, and are easy to slot into the Adam family. The derivation and notation are concise, and Algorithm 2 makes the change explicit.

### Minimal change, broad compatibility

Adam+ changes only the second-moment estimator and optionally injects tiny Gaussian noise, so it is easy to implement and to port to AdamW, AMSGrad, LAMB, and ADOPT. The appendix lists drop-in “+” variants and non-linear SNR-guided scalings, showing the idea composes with popular Adam-like optimizers.

**Weaknesses:**

### Larger optimizer state
Compared to Adam, Adam+ must keep $m_t$, $v_t$, and the extra EMA $w_t$, increasing state memory per parameter. While modest, this makes the footprint comparable to AMSGrad-style variants that also track a third buffer.

### Sensitivity in very low-variance regimes
When the centered variance becomes very small, the per-element scaling can explode without safeguards. The method relies on $\varepsilon$ or the optional noise injection to maintain a noise floor, which introduces additional knobs that may require tuning in some settings.

### Fairness versus per-optimizer retuning
The authors hold pipelines and most hyperparameters fixed to isolate the effect of centering, which is methodologically clear but may understate baselines that benefit from per-optimizer retuning. The paper emphasizes identical conditions, yet some optimizers are known to prefer different $\beta$ schedules or learning-rate rules.

### Possible lag under rapid non-stationarity
The mean gradient $w_t$ is a slow EMA using $\beta_2$. If the gradient distribution shifts abruptly, a slow $w_t$ could lag, briefly misestimating variance before adapting, which the paper partly mitigates with noise injection and shows empirically favorable behavior in RL but does not analyze formally.

**Questions:**

- Adam+ keeps $m_t$, $w_t$, and $v_t$, and injects noise per step. Did the authors measure the per-parameter state memory and step-time overhead versus Adam and AMSGrad on large models and mixed precision training, and can you share numbers for GPU RAM and wall-clock throughput?
- Algorithm 2 uses the same $\beta_2$ to form the slow mean $w_t$ and the centered variance $v_t$. Did the authors try decoupling these parameters differently, and does theory suggest an optimal relation between them?
- How sensitive is the scale of noise injection for the gradient estimate?

---

> ### Author Response · Authors · 2025-11-21
> **Part 1**
>
> We thank the reviewer for the feedback.
> >Larger optimizer state.
>
> It is true that Adam+ increases the memory footprint by introducing the slow mean estimator $w_t$. However, we argue that this modest cost is justified by the performance and theoretical benefits.
> As derived in Appendix H, the additional state $w_t$ is necessary to correctly center the second-order moment $v_t$. We prove that this specific addition allows Adam+ to converge to a smaller steady-state error ball compared to methods that track fewer states (like Adam) or use fast momentum for centering (like AdaBelief).
> Moreover, as the reviewer correctly notes, this footprint is comparable to widely adopted variants like AMSGrad.
> Since modern deep learning frameworks and hardware already support 3-state optimizers, Adam+ does not introduce unprecedented requirements.
>
> Lastly in our primary domain of RL, the memory consumption is overwhelmingly dominated by the experience replay buffer and environment simulation states.
> In this context, the marginal increase of adding one parameter vector for the optimizer ($3|\theta|$ vs $2|\theta|$) is negligible relative to the total system memory budget.
> >Sensitivity in very low-variance regimes[...]
>
> We have clarified the text to distinguish between the numerical safeguard and the optional exploration mechanism. Please notice that the primary safeguard against division-by-zero in low-variance regimes is the term $\epsilon$, not noise injection. This is a standard practice in widely adopted optimizers, such as in standard Adam, AdamW, RMSProp. Consequently, this is not an additional knob requiring new tuning
>
> In addition to that we clarified that the "optional noise injection" mentioned in the paper is strictly an exploration strategy for RL, analogous to $\epsilon$-greedy. In non-RL tasks (like the Crammed BERT or graph regression experiments reported in the paper), we did not use noise injection, and Adam+ remained stable and effective using only the standard $\epsilon$.
>
> >Fairness versus per-optimizer retuning
>
> For the majority of our experiments (MuJoCo, Atari, Crammed BERT, LSTM from AdaBelief), we did not pick arbitrary fixed hyperparameters. Instead, we utilized the specific configurations highly tuned for standard Adam/AdamW by the community or original authors. For RL we used the Stable-Baselines3 RL Zoo hyperparameters, which is result of the efforts of RL community tuning efforts aiming to maximize the performance of default optimizer (typically Adam) on a range of problems.
>
> For sequence modeling problems we used the exact configuration from the CrammedBERT and AdaBelief's LSTM repositories, which were optimized for AdamW and AdaBelief.
>
> Consequently, by holding these fixed, we are testing Adam+ as a replacement for the scenario scenario where the hyperparameters are suboptimal for our method but near-optimal for the baseline. The fact that Adam+ consistently outperforms the baseline under the baseline's own optimal conditions demonstrates algorithmic robustness and ease of use.
>
> For the molecular graph regression task, we performed the retuning for each optimizer. In the Table 12 and Table 13, we conducted an extensive grid search over learning rates and schedules for all optimizers. In this setting, Adam+ still exhibits performance gains.
>
> >Possible lag under rapid non-stationarity
>
> When the gradient changes abruptly, the variance $ \mathbb{E}[(g_t-w_t)^2]=\operatorname{Var}[g_t-w_t] +  \mathbb{E}[(g_t-w_t)]^2\approx \operatorname{Var}[g_t] + \mathbb{E}[(g_t-w_t)]^2$ can be decomposed as the variance in the steady-state (i.e., before the change) and the squared bias due to the abrupt change. We argue that this bias lag is beneficial because it automatically reduces the step-size in uncertain regions.
>
> >Algorithm 2 uses the same $\beta_2$ to form the slow mean $w_t$ and the centered variance [...]
>
> We chose to use the same $\beta_2$ for both the slow mean $w_t$ and the variance $v_t$ to maintain statistical consistency.
> This consistency is important for estimating the gradient SNR accurately.
> Adam+ update can be seen as an exponentially weighted version of Welford’s online algorithm for variance estimation, which can further support our argument.
> In statistics, a consistent estimator of the variance over a window of size $N \approx (1-\beta_2)^{-1}$ needs the mean to be estimated over the same time frame.
> If we were to separate them, for example, by using a faster decay for $w_t$, the mean estimator would closely follow high-frequency changes.
> This would remove the local signal we want to measure, turning the denominator into a high-pass filter instead of an actual variance estimator.
> By coupling the timescales, we ensure that $v_t$ measures the noise's dispersion relative to the steady gradient signal.

---

> > ### Author Response · Authors · 2025-11-21
> > **Part 2**
> >
> > > How sensitive is the scale of noise injection for the gradient estimate?
> >
> > Based on our experiments, we found the method to be empirically robust to the scale of noise injection within the range ($\sigma \leq 10^{-3}$). As reported in Table 4 (Appendix A), the sensitivity within the range of $10^{-5}$ to $10^{-3}$ is rather low.
> > Generally the exact tuning of $\sigma$ is not critical. To give some intuition, for the lower bound, the value of noise magnitude should be non-negligible compared to the numerical stability constant $\epsilon$ to provide a meaningful "noise floor".
> > For the upper bound it should not exceed the typical gradient magnitude to avoid conflating the noise with the learning signal. Crucially, we ended up employing noise injection solely for the RL tasks.
> > For other experiments, the SNR-centering mechanism provided improvements without any additional noise.
> > This confirms that noise injection is an optional, domain-specific exploration aid for RL.

---

> ### Author Response · Authors · 2025-11-21
> **Part 3**
>
> > Adam+ keeps $m_t$, $w_t$ and $v_t$, and injects noise per step. Did the authors measure the per-parameter state memory and step-time overhead versus Adam and AMSGrad on large models and mixed precision training, and can you share numbers for GPU RAM and wall-clock throughput?
>
> We benchmarked nanoGPT (GPT-2 124M) on a single NVIDIA A100 using mixed precision. For a fair algorithmic comparison, we tested our Python-based AdamW+ against the standard unfused AdamW and AMSGrad baselines. We adjusted the gradient accumulation steps to maintain a consistent compute load per optimizer update on the single-GPU setup. Also, we disabled the optional noise injection component for this benchmark, as our empirical analysis suggests this feature is primarily beneficial for RL.
>
> Runtime: As shown in Figure 4c in the main paper, the wall-clock training trajectories of AdamW and AdamW+ are very close resulting in a negligible <2% of runtime increase.
>
> For the memory we measured peak GPU memory usage during training: AdamW: 13.93 GB, AMSGrad: 14.41 GB (+0.48 GB) AdamW+: 15.95 GB (+2.02 GB). Theoretically, AdamW+ shares the exact same state footprint as AMSGrad ($3|\theta|$). The measured discrepancy (~1.5 GB) is due to temporary buffer allocations in our Python implementation. A standard fused kernel implementation would eliminate this difference, and would bring AdamW+’s and AMSGrad's memory usage significantly closer.

---

### Official Review · Reviewer_p2bK · 2025-11-09

**Soundness:** 3
**Presentation:** 3
**Contribution:** 3
**Rating:** 6
**Confidence:** 3

**Summary:**

This paper proposes Adam+, which replaces Adam’s second raw moment with the second central moment (variance), optionally with small noise injection, so updates become SNR-proportional, reducing over-attenuation in high-signal regimes and remaining drop-in compatible with other Adam-like optimizers. Validated across RL, sequence modeling, synthetic optimization, and graph regression, Adam+ shows consistently faster convergence, stronger final performance, and improved training stability.

**Strengths:**

- The paper is very well written and easy to follow.
- The proposed method simply replaces Adam’s second raw moment with the second central moment (variance), so that updates become SNR-proportional, mitigating over-attenuation in high-signal gradient regimes; the principle is clear and elegant. Moreover, this change is broadly applicable to various Adam-like optimizers, making it highly useful.
- Across diverse tasks, including RL and sequence modeling, the method consistently improves performance and stabilizes training, which supports strong confidence in the approach.

**Weaknesses:**

- While the proposed method, Adam+, keeps updates SNR-proportional by replacing the raw second moment with the centered second moment (variance) when estimating vt​ (thus avoiding unnecessary attenuation) the paper does not sufficiently explain how the second design choice, noise injection into the gradient sample, relates to Adam+. Please clarify whether noise injection is theoretically complementary to the SNR-centering (i.e., thanks to centering, adding noise does not break the updates and can be beneficial), or whether it is orthogonal (a regularization that would similarly help other optimizers as well).
- In experiments, results are not shown for noise injection with standard Adam/AdamW. Please add an ablation comparing Adam/AdamW (+ noise) under the same σ and settings to empirically disentangle whether the effect of noise injection is specific to Adam+ or independent of it.
- Figure 1 is hard to interpret from the caption alone. From the text, I understand it illustrates that under the proposed method the update is normalized by the noise standard deviation so that, even when noise grows in the late stage, the update becomes small and cautious. Please add a short explanation of this intent to the caption. Also, in the late stage label, shouldn’t γt1​ be γt2​?

**Questions:**

Please address the above concerns with concrete clarifications.

---

> ### Author Response · Authors · 2025-11-21
>
> We thank the reviewer for the positive evaluation of our work.
> > [...] Please clarify whether noise injection is theoretically complementary to the SNR-centering [...]
>
> Our results and analysis suggest that noise injection serves two purposes: first, as the reviewer points out, noise acts as a general regularizer that supports exploration in reinforcement learning.
>
> Secondly, and more importantly, noise injection is theoretically complementary to SNR-centering. In Adam, injected noise increases the uncentered second moment ($v_t \approx \mathbb{E}[g_t]^2 + \sigma^2 + \sigma^2_{inject}$), which can significantly reduce the step size when the signal is strong. In Adam+, the centering mechanism ($v_t \approx \sigma^2 + \sigma^2_{inject}$) separates the variance. This allows Adam+ to view the injected noise correctly as part of the noise floor, adjusting the step size accordingly.
>
> Therefore, while noise helps Adam with exploration, Adam+ can potentially work better with the noise by using it to adjust the adaptive learning rate.
>
> > [...] Please add a short explanation of this intent to the caption [Fig. 1]. Also, in the late stage label, shouldn’t $\gamma_{t1}$ be $\gamma_{t2}$?
>
> Yes, your understanding is correct. We have extended the caption based on your suggestion. We also corrected the typo with $\gamma_{t2}$.
>
> > In experiments, results are not shown for noise injection with standard Adam/AdamW. Please add an ablation comparing Adam/AdamW (+ noise) under the same σ and settings to empirically disentangle whether the effect of noise injection is specific to Adam+ or independent of it.
>
> Thank you for the suggestion. We have included the ablation in Appendix J. Based on the Hopper-v5 environment from the robotic control settings, and for the normally distributed noise with $\sigma\approx0.03$ and $\sigma=0.01$ (corresponding to -50 dB and -60 dB, respectively), we observed that the effect of noise injection is orthogonal to SNR-centering, meaning that Adam can also benefit from it.

---

### Author Response · Authors · 2025-12-02
**General comment summarizing all changes**

We thank the reviewers for their constructive feedback. We have uploaded a revised manuscript incorporating new theoretical analysis and empirical benchmarks. Below is a summary of the key updates

1. Theoretical convergence analysis (Appendix H), (reviewers RYRY, KLCS). For theoretical justification of our work, we added a convergence analysis on quadratic functions comparing Adam, AdaBelief, and Adam+. As a result, we prove that while AdaBelief achieves fast initial descent, Adam+ converges to a steady-state error ball of a smaller radius. This confirms our intuition that using a stable, slow-moving mean ($w_t$) for variance estimation allows for more precise step-size dampening near the optimum compared to using a fast-moving mean as in AdaBelief.
2. Large-scale computational benchmarks (reviewers PwsC, RYRY, KLCS). To address concerns regarding memory and throughput in large-scale settings, we conducted an end-to-end training benchmark on GPT-2 (124M) using Mixed Precision on NVIDIA A100s. We demonstrated that in this setting, the wall-clock overhead of AdamW+ is negligible compared to AdamW. We highlight that the memory footprint of Adam+ is structurally identical to AMSGrad ($3|\theta|$) and comparable to other modern $3|\theta|$ optimizers.
3. Ablation of noise injection (Appendix J), (reviewer p2bK). To disentangle the effects of SNR-centering vs. noise injection, we added an ablation study on Hopper-v5. We empirically confirm that noise injection is orthogonal to the optimizer choice (standard Adam can also benefit from it for exploration)
4. Expanded related work (reviewer KLCS). We expanded the discussion to include recent optimizers suggested by reviewers (AdaEMAMix, SOAP, Muon, Scion). We also clarified that our RL baselines utilize the community-standard, highly-tuned Stable-Baselines3 RL Zoo hyperparameters to ensure a fair comparison.

We believe these additions address the reviewers' questions regarding theory, scale, and mechanism.

---

### Meta-Review · Area_Chair_3oEV · 2026-01-07

**Summary:**

Most reviewers think the paper lacks sufficient novelty, rigorous theoretical justification, and comprehensive large-scale evaluation. Multiple reviewers found the contribution incremental relative to existing methods like AdaBelief and unconvinced by the rebuttal.

**Reviewer Concerns:**

While the authors added convergence analysis and clarified noise injection, concerns about novelty over AdaBelief, weak theoretical grounding, limited baseline comparisons, and insufficient scale of experiments remain unresolved. The claim of SNR-guided improvements lacks broader empirical and conceptual substantiation.

**Reviewer Scores:**

Reviewer p2bK (6-->6): Appreciated the simplicity and empirical gains. While clarifications were helpful, concerns on orthogonality of noise injection and broader applicability likely remain; score would likely stay the same.

Reviewer PwsC (4-->4): Despite detailed rebuttals, weaknesses around optimizer memory cost, stability under non-stationarity, and scale were not fully addressed; score likely unchanged.

Reviewer RYRY (2-->2): Major novelty concerns and unclear distinction from AdaBelief persist even after theoretical appendix updates; score likely unchanged.

Reviewer KLCS (2-->2): Acknowledged empirical breadth, but lack of stronger baselines, theoretical rigor, and larger-scale experiments remain limiting; score likely unchanged.

---

### Decision · Program_Chairs · 2026-01-26

Reject